# Precise immunofluorescence canceling for highly multiplexed imaging to capture specific cell states

Kosuke Tomimatsu[1,12], Takeru Fujii[1,12], Ryoma Bise[2], Kazufumi Hosoda [3], Yosuke Taniguchi [4], Hiroshi Ochiai [5], Hiroaki Ohishi [5], Kanta Ando[1], Ryoma Minami[1], Kaori Tanaka[1], Taro Tachibana[6], Seiichi Mori[7], Akihito Harada[1], Kazumitsu Maehara [1], Masao Nagasaki [8], Seiichi Uchida [2], Hiroshi Kimura [9], Masashi Narita [10,11] & Yasuyuki Ohkawa [1] ✉

Cell states are regulated by the response of signaling pathways to receptor ligand-binding and intercellular interactions. High-resolution imaging has been attempted to explore the dynamics of these processes and, recently, multiplexed imaging has profiled cell states by achieving a comprehensive acquisition of spatial protein information from cells. However, the specificity of antibodies is still compromised when visualizing activated signals. Here, we develop Precise Emission Canceling Antibodies (PECAbs) that have cleavable fluorescent labeling. PECAbs enable high-specificity sequential imaging using hundreds of antibodies, allowing for reconstruction of the spatiotemporal dynamics of signaling pathways. Additionally, combining this approach with seq-smFISH can effectively classify cells and identify their signal activation states in human tissue. Overall, the PECAb system can serve as a comprehensive platform for analyzing complex cell processes.

Cell states are dependent on various extrinsic factors for activating signal transduction. To understand cell states, responsive signaling pathways in cells have been profiled by analyzing post-translational modification of each signaling molecule. The approach to tracking individual cell responses has been analyzed using imaging-based techniques[1], but these techniques have limitations when simultaneously measuring the full spectrum of activated signaling pathways. Recent advances in single-cell omics technologies, such as single-cell

transcriptome and epigenome analysis can predict the responsive signaling pathway by gene set enrichment analysis (GSEA)[2,3]. However, extracting the dynamics of multiple signal activation in a cell is still challenging.

Multiplexed immunofluorescence has been developed to acquire information from multiple proteins and it has the potential to visualize signal activation in processes. For instance, the methods of ref. 4 and Sequential Indirect Immunofluorescence Imaging(4i)[5] strip

[1]Division of Transcriptomics, Medical Institute of Bioregulation, Kyushu University, 3-1-1 Maidashi, Higashi-ku, Fukuoka 812-0054, Japan. [2]Department of Advanced Information Technology, Kyushu University, 744 Motooka, Nishi-ku, Fukuoka 819-0395, Japan. [3]Ansanga Lab, 31, Yamadaoka, Suita, Osaka 565-0871, Japan. [4]Department of Medicinal Sciences, Faculty of Pharmaceutical Sciences, Kyushu University, 3-1-1 Maidashi, Higashi-ku, Fukuoka 812-0054, Japan. [5]Division of Gene Expression Dynamics, Medical Institute of Bioregulation, Kyushu University, 3-1-1 Maidashi, Higashi-ku, Fukuoka 812-0054, Japan. [6]Department of Chemistry and Bioengineering, Osaka Metropolitan University, Osaka 558-8585, Japan. [7]Cancer Precision Medicine Center, Japanese Foundation for Cancer Research, 3-8-31 Ariake, Koto-ku, Tokyo 135-8550, Japan. [8]Division of Biomedical Information Analysis, Medical Institute of Bioregulation, Kyushu University, 3-1-1 Maidashi, Higashi-ku, Fukuoka 812-0054, Japan. [9]Cell Biology Center, Institute of Innovative Research, Tokyo Institute of Technology, 4259 Nagatsuta, Midori-ku, Yokohama 226-8503, Japan. [10]Cancer Research UK Cambridge Institute, Li Ka Shing Center, University of Cambridge, Cambridge CB2 ORE, UK. [11]World Research Hub Initiative (WRHI), Institute of Innovative Research, Tokyo Institute of Technology, 4259 Nagatsuta, Midori-ku, Yokohama 226-8503, Japan. [12]These authors contributed equally: Kosuke Tomimatsu, Takeru Fujii. ✉e-mail: yohkawa@bioreg.kyushu-u.ac.jp

antibodies after indirect immunofluorescence, while Cyclic Immunofluorescence (CycIF) achieves inactivating fluorophores conjugated to antibodies using reactive oxygen species[6,7]. Although 4i and CycIF can detect activated signaling molecules, these methods use harsh conditions to remove fluorescence derived from fluorescence-labeled antibodies, which can potentially damage samples. This makes it difficult to achieve sufficient multiplexity to analyze cell states. Recently, antibodies conjugated with single-stranded DNA oligos that are detected with complementary single-stranded DNA probes in a fluorescence in situ hybridization (FISH)-based method have been established and widely adopted. After reacting a cocktail of oligo DNA-labeled antibodies with a sample, staining is achieved by sequentially reacting and stripping fluorescent DNA probes that are complementary to the sequence on each antibody[8–13]. However, the oligo DNA probes often non-specifically interact with nuclear proteins making it necessary to find optimal combinations of antibodies and oligo DNAs[9]. Therefore, oligo DNA-labeled antibodies have been limited to detecting cell surface antigens or the use of highly specific antibodies. Challenges therefore remain in the development of high-resolution, sequential staining techniques for visualizing signal transduction.

In this study, we developed precisely erasable fluorescent-dye labeled antibodies. The antibodies detected activated signals and nuclear proteins with specificity, and high multiplexity was achieved by gentle cleaving conditions that maintain the quality of the sample. Furthermore, through a reconstructive approach of single-cell analysis with automation, the activation of signaling pathways and changes in cell states were elucidated, enabling tissue-specific abnormalities to be identified.

## Results

### Development of Precise Emission-Canceling Antibodies (PECAbs)

To achieve sequential immunofluorescence with specificity, we designed an antibody attached to a fluorescent dye via a linker containing a disulfide bond (SS) (Fig. 1a). Antibodies were reacted with DBCO(dibenzylcyclooctyne)-SS-NHS(n-hydroxysuccinimide) ester and then with azide-modified fluorescent dyes to label the antibodies through the SS linker. We investigated the feasibility of cleaving fluorescent signals using antibodies against histone H3, an abundant basic nuclear protein. Anti-histone H3.1 primary antibodies were labeled with Alexa488 via an SS linker (SS-488) as described above.

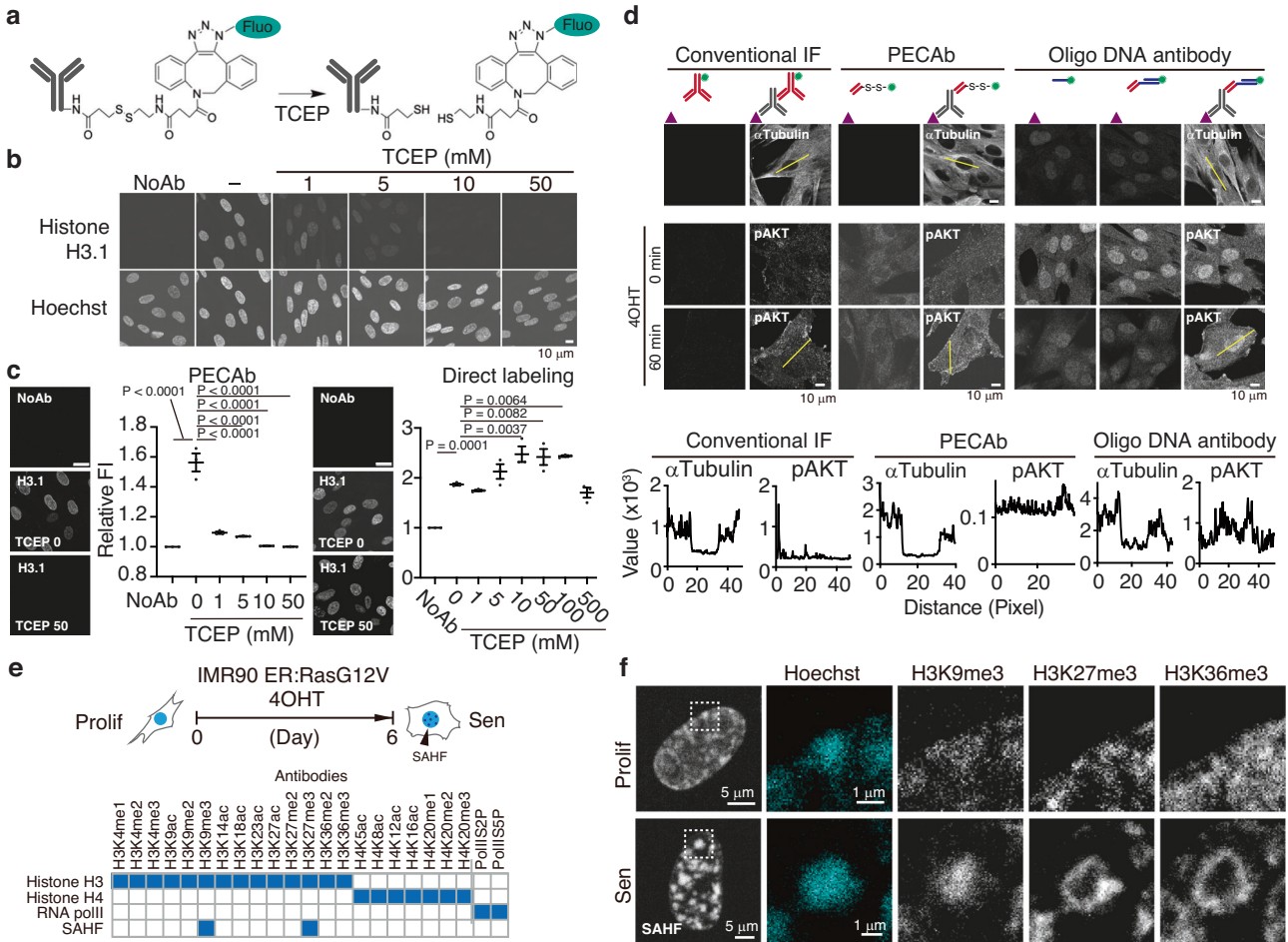

**Fig. 1 | Development of an erasable multiplexed-immunofluorescence method.**
**a** Schematic diagram of a fluorescence-cleavable antibody. **b**, **c** Immunofluorescence images of IMR90 cells stained with an anti-histone H3.1 PECAb or direct-labeled antibodies. The fluorescence signals were erased by TCEP. Scale bars, 10 μm in (**b**), 20 μm in (**c**). Quantified values were plotted in (**c**). Values are the mean ± SEM. Exact P values from one way ANOVA with Dunnett's multiple comparisons test (n = 3 biologically independent experiments) are indicated in the panels and Source Data. **d** Top panels, immunofluorescence images of IMR90 cells stained with indicated antibodies. Scale bars, 10 μm. Bottom panels, the profile of fluorescence intensity on the yellow line drawn on the immunostaining images using an indicated full antibody set.
**e** Schematics of SAHF staining. Top panel, induction of senescence. Bottom panel, PECAbs used to stain the IMR90 cell nuclei. Blue fill shows relevant features to the indicated antibody. **f** SeqIS images of IMR90 cells. Cells were sequentially stained with the antibodies. Areas within white dashed lines are magnified in the right panels. Scale bars for left: 5 μm, for magnified panels: 1 μm. Experiments in (**d**, **f**) were performed for 2 times and the representative images are shown. Source data are provided as a Source Data file.

Specific staining patterns were observed in nuclei of IMR90 human diploid fibroblasts. The fluorescence was erased after treatment for 30 min with the reducing agent, TCEP [tris(2-carboxyethyl)phosphine], in a concentration-dependent manner (Fig. 1b). The loss of fluorescence is attributed to the cleavage of both disulfide bonds in the linker and in the antibody molecules. To evaluate the efficiency of SS cleavage, we compared signals of anti-H3.1 antibodies directly labeled with Alexa 488 and with SS-488. TCEP treatment (10 mM) after immunofluorescence, decreased the fluorescence of SS-488 labeled antibodies to unstained levels, but no significant reduction in directly labeled antibodies was observed. These results indicate that fluorescence labeling of antibodies via SS linkers enables efficient cleaving of the fluorescent signal (Fig. 1c). These antibodies were named "Precise Emission Canceling Antibody" or PECAb. Oligo-DNA-labeled antibodies bind non-specifically in the nucleus[9,14]; therefore, we verified whether SS-fluorescent dye labeling affected these non-specific reactions. A comparison of the staining patterns was performed using PECAb and oligo DNA-labeled antibodies. Both α-Tubulin for cytoplasmic fiber structures and pAKT for signal transduction were immunostained by using PECAb as secondary antibody. Conversely, oligo DNA-labeled Fab and read-out probe alone showed non-specific signals in the nuclei, whereas it was minimal on the PECAb-Fab stained cells (Fig. 1d). We then determined the fluorescence intensity of line profiles drawn across the immunostaining images using images stained with the full antibody set. These line profiles indicated that the distribution of fluorescence intensity from PECAb staining more closely resembled that of conventional IF than the oligo-DNA antibody staining (Fig. 1d bottom). These results indicate that PECAb-based staining offers an advantage for high-specificity analysis of nuclear and signal transduction proteins. For other examples, 100 specific PECAb-stained images are shown in Supplementary Fig. 1. However, in multiplexed immunofluorescence, residual PECAbs might interfere with the detection of proteins aggregating within fine structures. To evaluate the specificity of PECAbs, we evaluated the staining of the specific nuclear structure, senescence-associated heterochromatin foci (SAHF)[15]. SAHFs are 1–2 µm microstructures that can be detected in the nucleus of IMR90 ER:Ras cells following treatment with 4OHT for 6 days to induce senescence (Fig. 1e)[16]. SAHF exhibits a hierarchical heterochromatin structure of core H3K9me3 surrounded by H3K27me3[17]. Therefore, we validated the simultaneous detection of these layered structures through Sequential Immunostaining (SeqIS) with PECAb using 23 antibodies, including 21 against histone modifications and two against PolII post-translational modifications (Fig. 1e). We observed hierarchical staining patterns of histone modifications within Hoechst-dense regions in senescent cells that were not observed in proliferative cells without RasG12V induction[17] (Fig. 1f, Supplementary Fig. 2). Clear signals were detected for all 23 antibodies, with H3K9me3 positioned at the SAHF core detected in the 4th cycle, surrounded by H3K27me3 detected in the 5th cycle, and H3K36me3, a euchromatin marker, detected in the 12th cycle, consistent with the previously reported pattern[17]. In addition, subnuclear quantitative analyses revealed four chromatin domain types based on epigenetic profiling (Supplementary Fig. 3a–d, described in Supplementary Note 1). These observations indicate minimal interference among antibodies. Taken together, these results indicate that the PECAbs can be used for a multiplexed immunofluorescence with higher specificity.

### Construction of sequential staining system using PECAbs
In sequential staining, data are acquired through multiple rounds of immunofluorescence and erasing[5–7]. To minimize processing differences among samples during a high number of rounds, we established an automated fluidics device, which enables the simultaneous processing of multiple samples (Fig .2a, Supplementary Fig. 4a).

To evaluate the staining efficacy and reproducibility of PECAbs on the device, duplicated IMR90 ER:Ras cells were independently cultured and separately seeded onto the same chip. SeqIS was performed for 21 cycles, resulting in the acquisition of 56-plex stained images (Fig. 2b, Supplementary Fig. 4b). The images indicated the following localization patterns: Lamin B1 on the nuclear envelope, α-Tubulin on cytoplasmic fiber structures, TOM20 on cytoplasmic mitochondrial structures, PML on nuclear spots, and GOLIM4 on the Golgi apparatus. (Fig. 2c). This result demonstrated the functionality of sequential staining with PECAbs using the automated device.

Quantitative analysis was performed to assess the technical reproducibility of SeqIS. Images were aligned using Hoechst staining as the reference, and erased images were used to subtract from stained images of the same cell for each cycle to compensate for the background (Fig. 2d, Supplementary Fig. 4c). Slight background accumulation was observed across the 21 cycles of staining (Fig. 2e, Supplementary Fig. 4d). Data were acquired from 1531 to 1520 cells for each replicate (Supplementary Fig. 4e), and comparison of the median between replicates revealed a remarkably high correlation (Fig. 2f). Uniform Manifold Approximation and Projection (UMAP) indicated consistent clustering between replicates (Fig. 2g). These results demonstrated that SeqIS on the automated device produced a dataset with high reproducibility.

### Profiling cell state dynamics using quantitative data from SeqIS
Single-cell data of sufficient throughput and depth allows cell states to be classified and dynamics to be extracted. Therefore, we assessed whether the cell state dynamics can be visualized using quantitative SeqIS. The IMR90 oncogene-induced senescence model (Fig. 3a), involves dynamic changes in the cell state and signal transduction. The cells were fixed in the following time-points after RasG12V induction, (i) Proliferative: day 0, (ii) Transition: day 3", and (iii)Senescent: day 6, and used for the SeqIS staining. We employed 206 antibodies to analyze the DNA damage response, apoptosis, senescence, and signaling related to senescence (Fig. 3b, Supplementary Fig. 5a). To assess whether the sequential staining with 206 antibodies captured changes in cell states, representative plex staining images for 56 of the antibodies were visually inspected at day 0 and day 6 (Fig. 3c). Comparison of images from day 0 and day 6 samples indicated the features of senescence, such as increased HMGA1 and decreased Lamin B1 staining[18,19]. Furthermore, cytoplasmic compartmentalization of autophagy-related protein, SQSTM1, and nuclear localization of pATF2 were observed on day 6, revealing changes in other proteins associated with senescence.

Next, images were quantified to classify cell states. We assessed 9002 cells from duplicate datasets on days 0, 3, and 6 (Supplementary Fig. 5b). Although background signals were accumulated beyond 20 cycles in channels 488 and 594 (Supplementary Fig. 5c), the data were quantified and offset using erased data. To evaluate the effect of accumulated antibodies on staining in later cycles, staining images of Lamin A in the 56th cycle, H3.1 in the 54th cycle, and α-Tubulin in the 73rd cycle were compared with their 1st cycle images. In this 73-cycle experiment, antibody accumulation did not prevent the detection of structural features of these proteins in the cell (Supplementary Fig. 5d). Additionally, to evaluate the relevance of epitope damage to antibody staining patterns, IMR90 cells were treated with 10 mM TCEP 50 times and stained with anti-Lamin A (polyclonal) and anti-α-Tubulin (monoclonal) antibodies. Cells were segmented and the fluorescence intensity (FI) was quantified. Although the FI signal using monoclonal anti-α-Tubulin was not decreased, signal using polyclonal anti-Lamin A, which is a mixture of antibodies of different quality, was decreased by TCEP treatment (Supplementary Fig. 5e). These results indicate that the effect of epitope damage on staining depends on the quality of the antibody. Monoclonal antibodies were used to prepare most PECAbs.

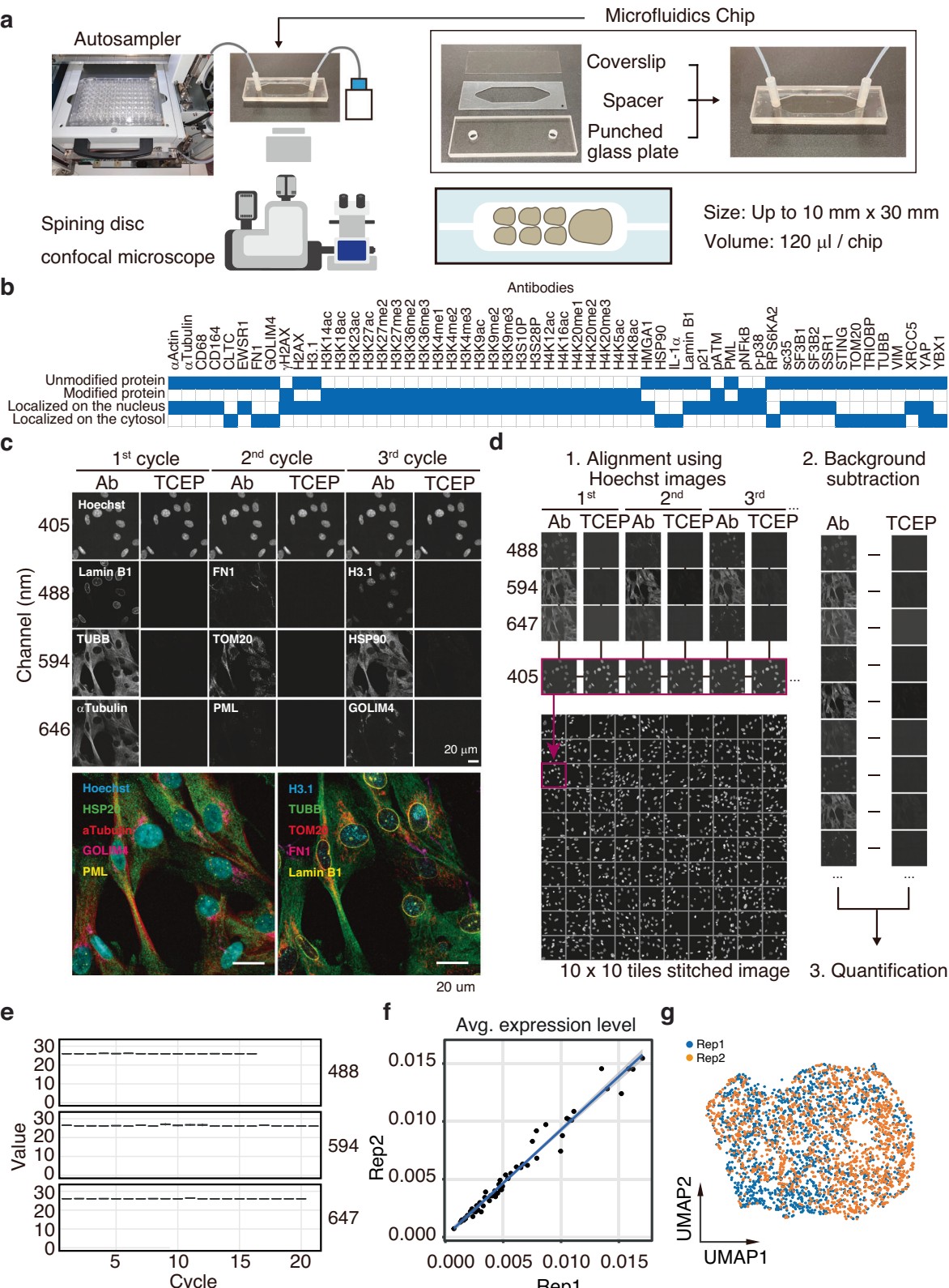

**Fig. 2 | Establishment of a sequential staining system. a** Schematic diagram of the automated device. **b** PECAbs used for the SeqIS experiments. Blue fill shows relevant features to the indicated antibody. Experiments were performed for 2 times and representative images are shown in (**c, d**), analyzed data are shown in (**e–g**). **c** Representative SeqIS images of IMR90 cells. Sequential SeqIS was performed on the automated device using antibodies indicated in (**b**). Scale bars, 20 μm. **d** Analysis workflow. 1) Image tiles were aligned on a stitched image based on Hoechst staining. 2) Cleaved signals were subtracted from stained images. 3) Cells were detected and fluorescence intensity in each area was quantified. **e** Quantified retained signals. The medians, 25th–75th percentiles, and 1.5 × interquartile range are employed to draw the box plots. **f** Correlation of quantified values between replications. Averages of the expression levels per antibody are plotted. **g** UMAP visualization to show the reproducibility.

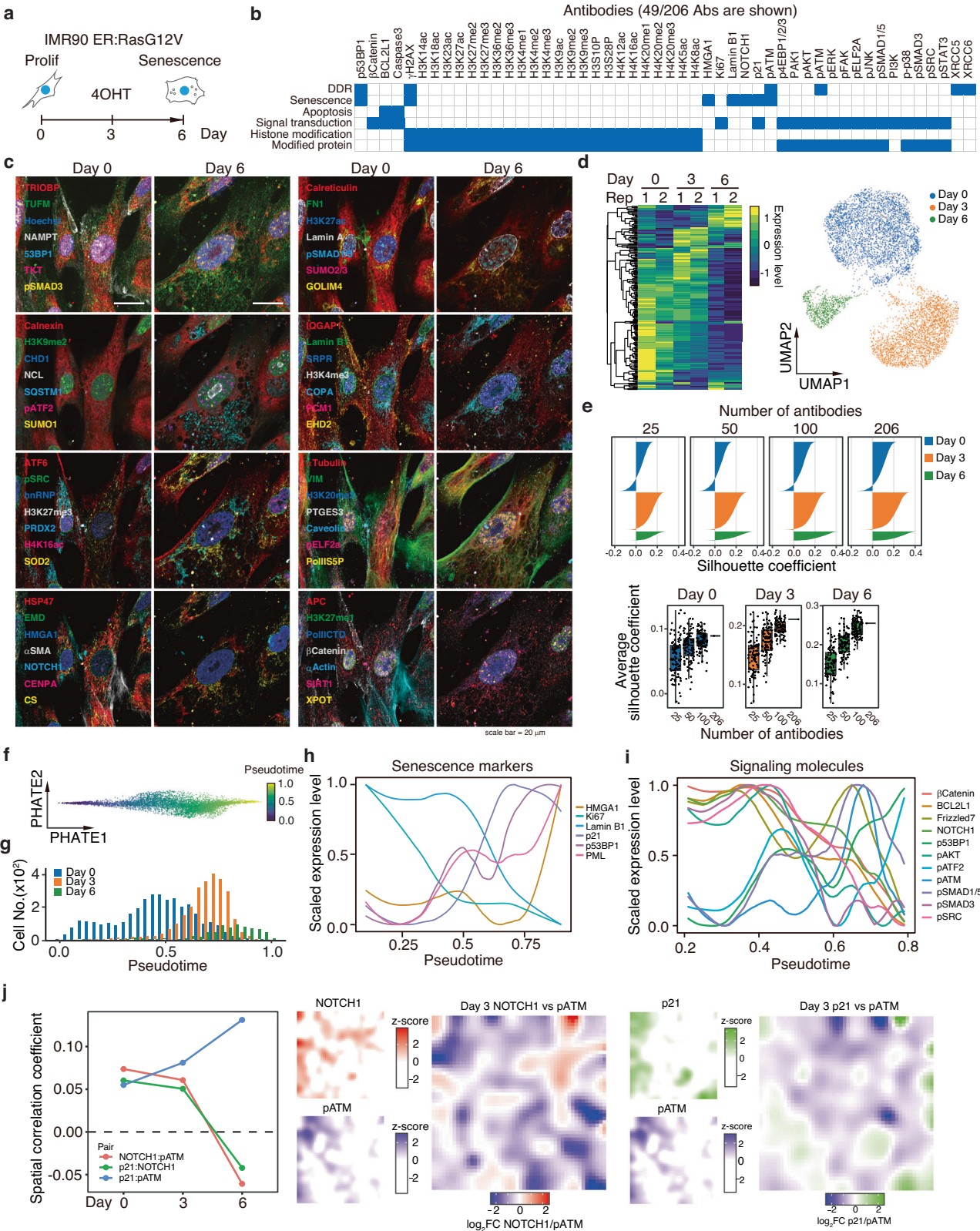

The staining signals were quantified to profile cell states. Comparison of quantitative data for days 0, 3, and 6, showed different profile for each day (Fig. 3d left). Moreover, UMAP revealed a clear separation of cell states for days 0, 3, and 6 (Fig. 3d right), with reproducibility (Supplementary Fig. 6a). These results show that quantitative values from SeqIS images can effectively separate cell states.

To examine the viability of using 206 antibodies to classify the cell state, the influence of antibody number on separation performance was assessed. Silhouette coefficients were calculated to compare the number of antibodies and cohesion within clusters. The results indicated that an increase in coefficients was dependent on the number of antibodies for all samples (Fig. 3e top). Additionally, randomly selected antibody sets of 25, 50, and 100 antibodies were created, and

**Fig. 3 | Analyzing cell state dynamics with a 206-plexed SeqIS dataset. a** IMR90 ER:Ras cells were treated with 100 nM 4OHT to induce senescence were picked on days 0, 3, and 6, then used for SeqIS. **b** PECAbs used for the SeqIS. Blue fill shows relevant features to the indicated antibody. Experiments were performed for 2 times and representative images are shown in (**c**), analyzed data are shown in (**d**–**j**). **c** SeqIS images of IMR90 ER:Ras cells. Representative 56-plexed images from day 0 and 6 samples are shown. Scale bars, 20 μm. **d** Left panel, heatmap showing average protein levels. Right panel, UMAP of quantified cells. **e** Silhouette analysis of the subsampled dataset. Upper panel, silhouette coefficients at each number of antibodies. 25, 50, or 100 antibodies were randomly extracted. Lower panel, box plots

indicate the average silhouette coefficient using 25, 50, 100, and 206 antibodies. The analysis was repeated 100 times using different random numbers for each antibody set. Boxes, 25th–75th percentiles; center, median whiskers, 1.5 × inter-quartile range. **f** PHATE embedding using senescence-associated proteins and inferred pseudotime. **g** Histogram of cells in pseudotime. **h** Dynamics of senescence-associated protein levels in pseudotime. **i** Dynamics of signal transduction during pseudotime. **j** Spatial expression patterns of NOTCH1, pATM, and p21 on day 3. The stitched image was divided onto a 50 × 50 grid. Protein levels were averaged for each grid. Source data are provided as a Source Data file.

silhouette coefficients were calculated 100 times. Analysis of Day 0 and 3 samples individually using 25 antibodies demonstrated separation performance exceeded that of the 206-plex (Fig. 3e bottom). However, when days 0, 3, and 6 were analyzed together, the 100 antibodies surpassed the score of 206-plex in only 3 out of 100 trials. These results show that a smaller number of antibodies can separate cell states by selection, but that antibodies above 200 can provide stable separation to unbiased.

To evaluate the dynamics of cell state changes, the quantified IMR90 cell dataset for senescence days 0, 3, and 6 was embedded using PHATE, and the pseudotime was constructed using PHATE coordinates (Fig. 3f). We found a correlation between the time samples were collected and pseudotime progression (Fig. 3g). Senescence markers were then plotted along the pseudotime. Factors with low quantitative values and exaggerated variability because of scaling were removed by visual inspection (Supplementary Fig. 6b, c). With the progression of pseudotime, characteristic features of senescence were observed, including decreased levels of Lamin B1 and increased levels of p21 and HMGA1. This demonstrated that pseudotime reconstructed from the data reproduced the temporal changes of cellular senescence (Fig. 3h).

With pseudotime accurately representing temporal changes in cell states, the possibility of visualizing the signal dynamics was explored by plotting signaling molecules along pseudotime (Fig. 3i). Consistent with previous studies[20], NOTCH1 was decreased around pseudotime 0.7, corresponding to the peak of day 3 cells. This was accompanied by a decrease in the level of pSMAD3, a downstream component of TGFβ signaling. Conversely, upon NOTCH reduction, transient activation of ATM and AKT was observed. NOTCH1 signaling has been reported to directly suppress the DNA damage response[21]; therefore, we analyzed spatial correlation of NOTCH1 and pATM. NOTCH1 and pATM were spatially exclusive on day 3 and day 6 but not on day 0, and the correlation between pATM and senescence marker, p21, increased with time (Fig. 3j). This suggests that the reduction of NOTCH might be associated with pATM activation during senescence. Taken together, these results demonstrated that analysis of signal transduction dynamics involved in cell state changes can be conducted using pseudotime reconstructed from SeqIS staining data.

### Assessment of signal transduction activation by tracking the dynamic changes

We assessed time resolution of cell state dynamics using SeqIS, which simultaneously acquires information about cell state and activation of signal transduction. We employed an epithelial-to-mesenchymal transition (EMT) system using A549 cells, where signal transduction is rapidly activated by adding the ligand, TGFβ[22]. To examine the time point of altered gene expression during EMT, A549 cells were treated with TGFβ (Fig. 4a). Cells were then harvested at 0, 0.5, 12, 24, 48, and 96 h, and analyzed by scRNAseq. Bulk correlation and single-cell analysis using UMAP showed that gene expression changed from 12 h after TGFβ treatment. Cells treated with TGFβ for 0.5 h showed weak genome-wide transcriptional changes from transcription at 0 h (Fig. 4b). We then performed SeqIS for samples at 0, 0.5, 24, and 96 h using a 27-plex PECAb set that included signaling activation and EMT

markers (Fig. 4c). Images were obtained and quantified to profile cell states. Data from duplicate samples at 0, 0.5, 24, and 96 h resulted in the detection of 36,955 cells (Supplementary Fig. 7a, b). Phosphorylated protein values, indicating signal transduction activation, were increased at 0.5 h, along with pSMAD3 activation (Fig. 4d, e). Non-phosphorylated protein values, such as those for EMT markers (N-cadherin, VIM and FN1), showed changes after 24 h (Fig. 4d, e). Furthermore, clustering by UMAP separated cell states at 0, 24, and 96 h, with 0 h and 0.5 h also being distinguished (Fig. 4f). These results were consistent with the silhouette results in Fig. 3e and demonstrated that selection of antibody sets that detect signal transduction activation allows for the separation of short-term cell state changes, which is challenging using conventional protein or RNA datasets. Following the separation of cell states between 0 and 0.5 h, the ability to visualize their dynamic changes using the reconstructed data system was verified (Fig. 4g, h). To further analyze the dynamics of signal transduction before cell state changes, activated signal transduction molecules were plotted against pseudotime (Fig. 4i). Factors with low quantitative values and exaggerated variability because of scaling were removed through visual inspection (Supplementary Fig. 7c, d). Following the addition of TGFβ, increased levels of downstream pSMAD3 were observed as pseudotime progressed. Furthermore, an increase in the level of pSMAD1/5 was observed after the increase in pSMAD3 level, activation of pERK and pFAK, and an early transient decrease in the level of pELF2a. Compared with stained cells in panel i in Fig. 4i, those in panel ii show no change in pSMAD1/5 and a decrease in pELF2, and those in panel iii exhibited strong pSMAD3 staining (Fig. 4j). These results demonstrated that SeqIS can capture the dynamics of signal transduction that precede cell state changes.

### Expansion of analysis through Integration of PECAb with Seq-smFISH

The targets that can be analyzed by sequential PECAb staining are limited by the availability of reliable antibodies. To address this, we established a multimodal approach by combining SeqIS with single-cell spatial transcriptomics single-molecule RNA fluorescent in situ hybridization (Seq-smFISH).

SeqIS requires staining, followed by erasing with a reducing agent. This may cause sample damage and loss of RNA. To address this, we first evaluated sample preservation through repeated cleaving treatment using the IMR90 cell line. Clear nuclear and mitochondrial structures were observed after repeated TCEP treatment, similar to those observed after repeated PBS treatment, whereas $H_2O_2$ quenching used for CycIF[6] resulted in the loss of organelle structures (Fig. 5a). The 3 M GCUrea-treated cells used for 4i[5] could not be prepared for electron microscopy analysis because the cells were degraded (Supplementary Fig. 8a). TCEP erasing preserved over 70% of RNA in cells compared with that preserved after PBS treatment (Fig. 5b). Consistent with RNA preservation, 70% of spot numbers were maintained in Seq-smFISH experiment using the TCEP treated sample (Supplementary Fig. 8b). These results indicate that repeated TCEP cleaving resulted in superior RNA preservation compared with other erasing conditions.

Following SeqIS, we then performed Seq-smFISH. To reduce background signals that were stronger than single-molecule RNA

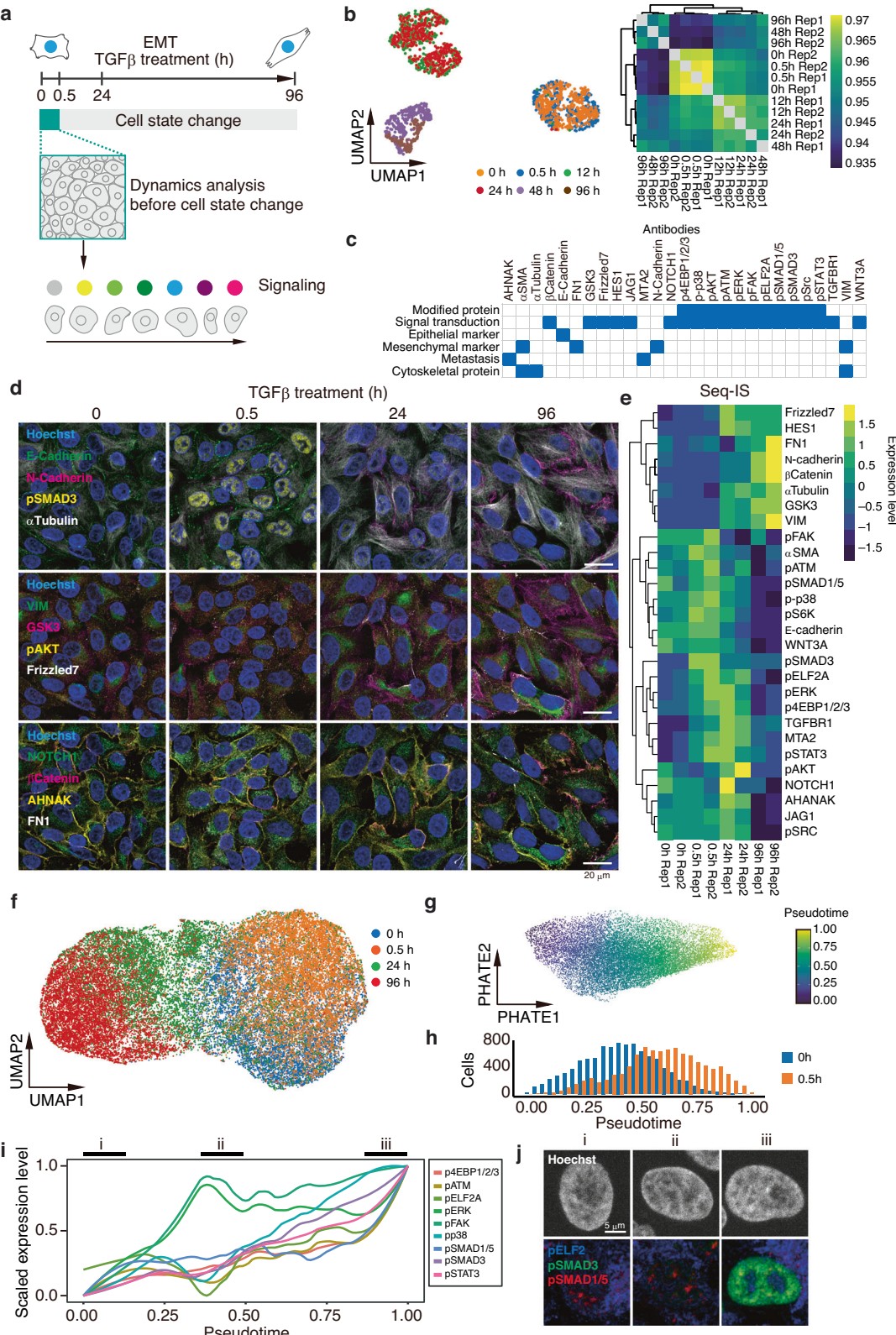

spots, protein clearing was necessary after the SeqIS (Fig. 5c). How-ever, it is necessary to retrieve samples from the microfluidics chip to perform protein-clearing experiments. To overcome this, we fixed a detachable thin film to the chip spacer, allowing samples to be peeled off the chip after sequential staining (Fig. 5d). Protein clearing was then performed after the sample was embedded in hydrogel. The detached chip can then be adhered to coverslips again using double-sided tape

and subjected to Seq-smFISH. SeqIS was performed for three rounds (9-plex), followed by 17 rounds of Seq-smFISH (34 genes) on biologi-cally duplicated samples. Using a public IMR90 cell RNA-seq dataset[20], primary probes were designed to detect randomly selected genes expressed at high, medium, and low levels (Supplementary Fig. 8c). SeqIS images revealed characteristic signal patterns, such as signals at the nuclear periphery indicative of H3K9me3, signals from inactivated

**Fig. 4 | Tracking signal transduction dynamics in a short time scale. a** Schematic diagram of the analysis of signal transduction dynamics. A549 cells were treated with TGFβ to induce EMT via activation of the SMAD3 pathway. Focusing on TGFβ for 0–30 min, signal dynamics were tracked in pseudotime. **b** Clustering of the cell state using sc-RNA seq datasets. A549 cells were treated with TGFβ for 0, 0.5, 12, 24, 48, and 96 h followed by transcriptome analysis. Left panel, UMAP projections of the transcriptome data. Right panel, correlations of the transcriptome datasets. **c** Antibodies used for the SeqIS experiment. Blue fill shows relevant features to the indicated antibody. Experiments were performed for 2 times and representative

images are shown in (**d**, **j**). **d** Immunofluorescence images of A549 cells. Representative 13-plexed images from 0, 0.5, 24, and 96 h samples are shown. Scale bars, 20 μm. **e** Heatmap showing average protein levels for each stitched image. **f** UMAP projection using the whole dataset. **g** PHATE embedding using quantified datasets at 0 and 0.5 h and inferred pseudotime. **h** Histogram of cells with pseudotime. Cells treated with TGFβ for 0 and 0.5 h were used. **i** Dynamics of signal transductions during pseudotime. **j** Representative immunofluorescence images of A549 cells at the indicated point in (**i**). Scale Bar = 25 μm.

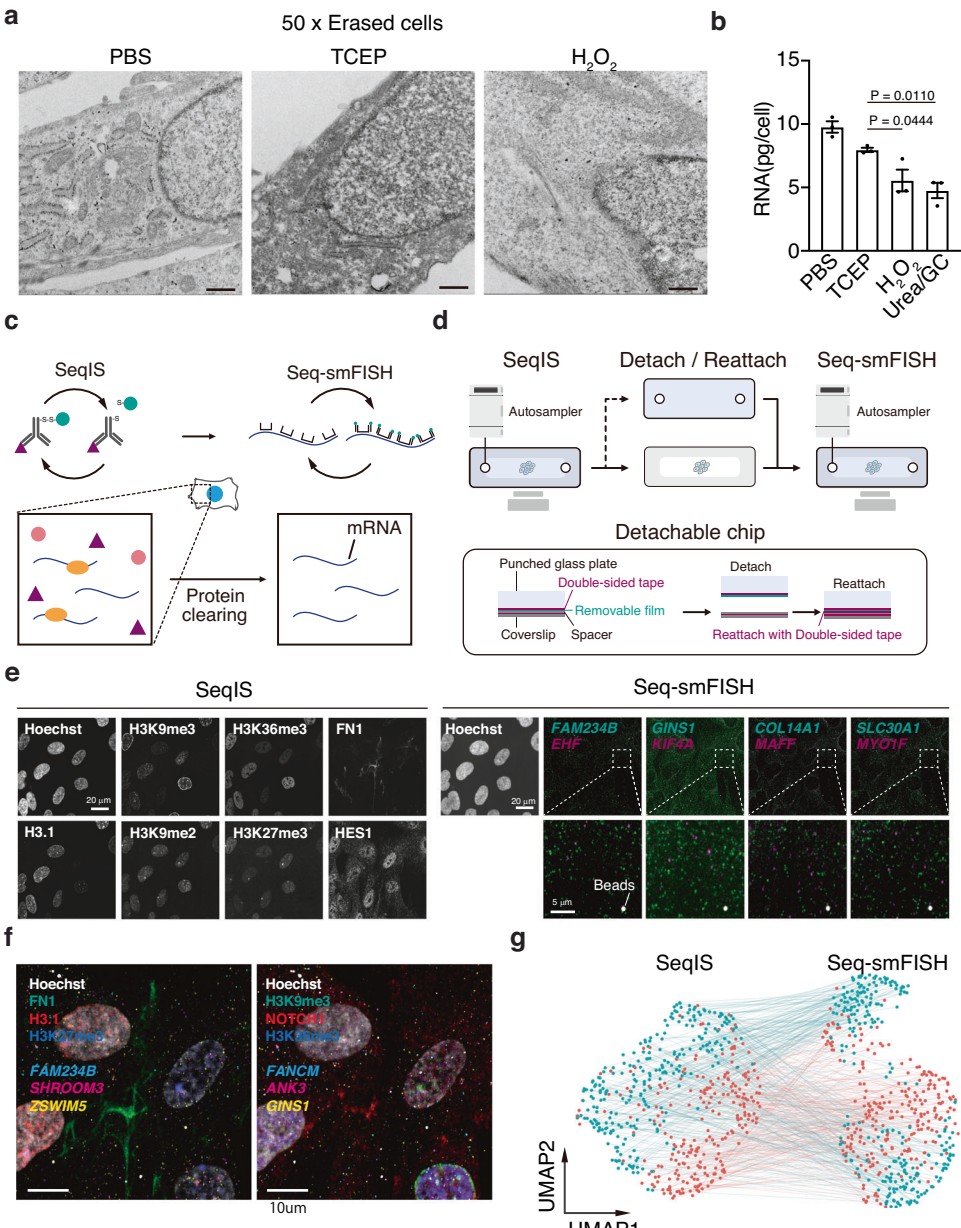

**Fig. 5 | Simultaneous analysis of proteins and RNAs. a** Electron microscopy images of IMR90 cells. Cells were treated 50 times with the indicated reagents before observation. Scale bar, 500 nm. **b** Preservability of RNA. Values are the mean ± SEM. Exact *P* values from one way ANOVA with Tukey's multiple comparisons test (*n* = 3 biologically independent samples) are indicated in the panel and Source Data. **c** Schematic illustration of multi-omics experiments. Proteins in SeqIS stained samples were cleared and the sample was subjected to Seq-smFISH. Experiments were performed for 2 times and representative images are shown in

(**e**, **f**). **d** Schematic illustration of the detachable chip for multi-omics experiments. **e** SeqIS and Seq-smFISH images of IMR90 cells. Left panel, SeqIS-stained IMR90 cells. Right panel, Seq-smFISH stained IMR90 cells after the SeqIS. Dash-lined squares are magnified in the lower panels. Scale bars, 20 μm in SeqIS and Seq-smFISH panels, 5 μm in the bottom images. **f** Merged images shown in (**e**). Scale bar, 10 μm. **g** UMAP projection of Seq-smFISH and SeqIS. Each dataset is projected respectively (*n* = 2). The points derived from the same cell are linked. Source data are provided as a Source Data file.

X chromosomes representative of H3K27me3, and fiber structures associated with FN1 (Fig. 5e). Subsequently, Seq-smFISH was performed and revealed different patterns of spots for the detected genes and acquired protein and RNA data from the same sample (Fig. 5e, f). For quantification of Seq-smFISH, although background noise was minimized by protein clearing (Supplementary Fig. 8d, unstained), the ReadOut probe signal was different from that of RNA spots (Supplementary Fig. 8d, center). Therefore, the RNA spots were determined by adjusting the brightness and the maximum and minimum pixel size parameters to distinguish noise from RNA spots (Supplementary Fig. 8d right). UMAP projection was applied to the simultaneously acquired immunofluorescence and FISH data for single cells and both immunofluorescence and FISH showed reproducibility between replicates (Fig. 5g). These results demonstrated that integration of SeqIS and Seq-smFISH can enable the simultaneous analysis of factors not detected by antibodies.

### Detection of cell activation states in tissues

To demonstrate the ability to detect cell state changes at the single-cell level in tissues, we evaluated a spatial multimodal approach combining SeqIS with Seq-smFISH (Fig. 6a). A uterine carcinosarcoma, composed of mixed carcinoma and sarcoma cells[23], was chosen for analysis. To investigate differences between the carcinoma and sarcoma components of the tissue, samples with different ratios of carcinoma and sarcoma cells (carcinoma:sarcoma = 30:70, 10:90, and 0:100) were first analyzed by RNA-seq (Fig. 6b). Correlation between samples based on gene expression profiling was decreased depending on the ratio of carcinoma components (Fig. 6c). Samples with 30% and 0% carcinoma components were then selected to extract the differentially expressed genes, and 220 genes were extracted to design probes for seq-smFISH (Supplementary Fig. 9a). Using these gene sets, GSEA was performed using the Molecular Signatures Database (MSigDB) to interpret their biological roles. This revealed a prominent EMT signature (adjusted $P$ value = 0.003241) (Fig. 6d, Supplementary Fig 9b), indicating EMT in the carcinosarcoma tissue. Subsequently, Seq-smFISH was performed for carcinoma 30%/sarcoma 70% samples using the identified gene set (Supplementary Fig. 9c). Based on the quantitative Seq-smFISH data, cells were classified cells into five states (Fig. 6e). To examine the spatial localization by state, clustered cells were mapped in space using tissue coordinates (Fig. 6f). According to silhouette analysis, Clusters 1 and 2 were more prone to aggregating cells with similar profiles in the tissue compared with Clusters 3, 4, and 5 (Supplementary Fig. 9d). Comparing the spatial expression of EMT-related gene sets with the classified clusters, only *EFEMP2* exhibited similar distribution patterns with Cluster 2; no other genes had distinctive distribution of spatial expression patterns (Fig. 6g).

To examine in detail the characteristics of the separated clusters, SeqIS was performed using markers for epithelial and mesenchymal cells and with activated signal transductions (Fig. 6h). Cluster 2 exhibited high levels of the epithelial marker E-cadherin, while Clusters 3, 4, and 5 showed high levels of the mesenchymal markers N-cadherin, FN1, and VIM (Fig. 6i). Cluster 1 exhibited lower E-cadherin levels compared with Cluster 2 and showed activation of signal transduction pathways that induce EMT, including pSMAD3 (Fig. 6i). The proximity of Cluster 1 to Cluster 2 suggested that EMT was activated in Cluster 1. Based on these results, we classified Cluster 2 as epithelial, Clusters 3, 4, and 5 as mesenchymal, and Cluster 1 as an EMT group (Fig. 6j, Supplementary Fig. 9e). The expression of signal transduction and epithelial and mesenchymal markers was profiled in regions with high occupancy of Clusters 1 and 2 (dashed white outline) (Fig. 6j). In Cluster 1, regions with activated signal transduction, such as pSMAD1/5, pSTAT3, and p-p38, exhibited activation in areas with E-cadherin expression (Fig. 6k, Supplementary Fig. 9f). The levels of *ZEB1*, a transcription factor that activates EMT, were higher in the regions with activated signal transduction (Fig. 6l). EMT induction suppresses

apoptosis, but the expression of the apoptosis-related gene, *DAP3*, was low in these regions (Fig. 6l). These results indicate that EMT is activated in Cluster 1 adjacent to the epithelial cell Cluster 2. Collectively, the spatial multi-omics integration of SeqIS and Seq-smFISH detected activated cell state changes in tissue.

## Discussion

In this study, we developed PECAbs, antibodies labeled with fluorescent molecules via SS linkers that can be efficiently cleaved of their fluorescent signals. By minimizing non-specific binding within the nucleus and sample damage, we achieved high-specificity and high-multiplexed imaging of molecules, including signal transduction components, in cell lines and tissues. With our automated system, the PECAb system has the potential to become a standard technology in the evolving field of spatial omics.

We performed sequential staining using up to 206 PECAbs. This highly-multiplexed staining approach has the potential for antibody interference with subsequent sample epitopes. Other approaches to highly multiplexed staining, in which all antibodies react together, such as CODEX[8,9] or ImmunoSABER[10], may lead to different types of interference compared with sequential staining methods. Interference in the PECAb system may be caused by antibody accumulation in later staining cycles. Therefore, we carefully chose the antibody staining order with low staining efficiency antibodies for the initial cycles and high efficiency antibodies for the latter cycles to enable sequential detection of specific staining patterns. For example, α-Tubulin staining in the 73rd cycle showed the same fiber-like structure as in the 1st cycle (Supplementary Fig. 5d). While interference from individual immunofluorescence may occur with similar frequencies across cells and targeted molecules, this would have minimal effect on the relative quantitation-based analysis conducted in this study. However, to obtain absolute quantitative data, alternative approaches such as efficiently removing antibodies, in addition to cleaving fluorophores, might be necessary.

In this study, we used antibody panels that mainly represented signal transduction activations, but direct measurements of cell-cell interactions will require an expanded variety of antibodies targeting ligands and receptors. Current spatial omics technologies determine cell types and analyze their spatial distribution[24,25], while methods for direct analysis of cell-cell interactions are primarily based on the expression patterns of ligand and receptor RNAs[26]. Measurements of actual cell-cell communication remain challenging. While the PECAb-based system enables specific and potentially direct measurements, higher-resolution imaging in the current system may lead to prolonged imaging times and reduced throughput. We expect that improvements in imaging technology will help overcome this limitation.

We have achieved a spatial multi-omics approach combining PECAbs and RNA Seq-smFISH, enabling the simultaneous acquisition of protein and RNA expression datasets with spatial information. Transcribed RNAs are not only translated to proteins but also contribute to the regulation of gene expression by interacting with proteins. PECAbs can detect protein localization with high specificity and, therefore, have potential application in the analysis of RNA-protein complex dynamics during cell state changes.

## Methods

### Ethical statement

Ethical approval was obtained from the internal review boards of the Japanese Foundation for Cancer Research (IRB-ID: M22116-00) for experiments using human tissues. Recruited patients provided written informed consent for their samples to be used in research. There was no participant compensation in this study.

All animal procedures were conducted in accordance with the Guidelines for the Care and Use of Laboratory Animals and were

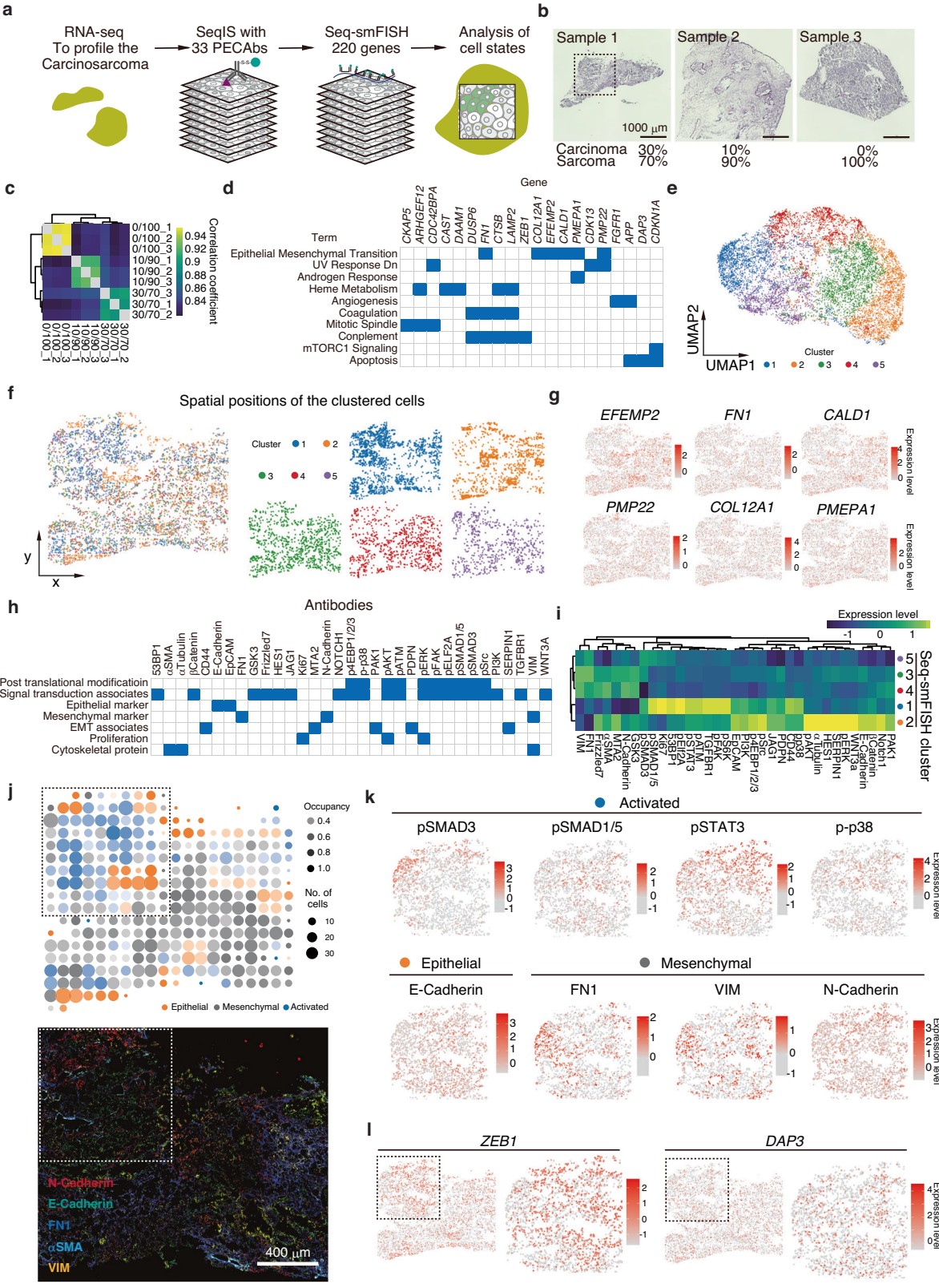

approved by the Institutional Animal Care and Use Committee (IACUC) at Kyushu University (Approved ID: A23-288-1).

## Antibodies

Antibodies used in this study are listed in Supplementary Data 1. BSA-free antibodies were purchased to facilitate the buffer exchange step.

## Cell culture

IMR90 ER:RAS (oncogene HRAS$^{G12V}$ fused to the estrogen receptor (ER) ligand-binding domain; ER:RAS) cells[16] were kindly provided by Masashi Narita (CRUK-CI). The cells were cultured in phenol-red-free Dulbecco's modified Eagle's medium (DMEM) (Gibco, 31053-028) with 10% fetal bovine serum under physiological 5% O$_2$ conditions. To induce senescence, the cells were treated with 100 nM 4-hydroxy

**Fig. 6 | Analysis of activated cell states in carcinosarcoma tissue. a** Schematic diagram of carcinosarcoma tissue analysis. Tissue RNA-seq data were used to design Seq-smFISH probes. Tissue was stained with 33 PECAbs followed by Seq-smFISH for 220 genes. All experiments were performed at least 2 times. **b** Profiling carcinosarcoma tissues. Sample 1, 2, and 3 were derived from different patients, respectively. The dashed-lined square shows the area analyzed by SeqIS and Seq-smFISH. Scale bar, 1000 μm. **c** Gene expression profiling to select carcinoma-associated genes. **d** Gene Set Enrichment Analysis using selected genes analyzed by seq-smFISH on MsigDB. **e** UMAP projection using the 220 Seq-smFISH genes. **f** Left panel, spatial mapping of clustered cells. Right panel, spatial positions of cells in individual clusters. **g** Spatial expressions of indicated genes. **h** PECAbs used in the carcinosarcoma analysis. Blue fill shows relevant features to the indicated antibody. **i** Average profiling of protein levels in each of the clusters. **j** Upper panel, the majority cell state groups in each 20 × 20 grid. Occupancy is the fraction of the dominant cell state in the grid. Lower panel, merged image of EMT markers. Scale bar, 400 μm. **k** Spatial expression profiles of indicated proteins. **l** Spatial expression of *ZEB1* and *DAP3* from Seq-smFISH analysis.

tamoxifen(4OHT) and cultured for 6 days. A549 (CCL-185) cells were obtained from the American Type Culture Collection (ATCC) and cultured in DMEM (Gibco, C11995500BT) with 10% fetal bovine serum. To induce EMT, the cells were seeded 24 h before induction to reach ~70% confluence and then treated with 4 ng/mL TGF-β1 (Pepro Tech, 100-21). The medium containing TGF-β1 was replaced every other day.

### Human tissues
Tumor samples were obtained from three female patients (55, 56, and 61 years old) to analyze uterus carcinosarcoma through standard surgical procedures. Patients had not received chemotherapy or radiation therapy prior to surgery. Pathological diagnosis of carcinosarcoma was performed by at least two independent gynecological pathologists. Tissues were snap-frozen in liquid nitrogen and stored at −145 °C.

### Mouse Tissues
C57BL/6 N (male, 17 weeks) mouse was purchased from the Jackson Laboratory. The tibialis anterior (TA) muscles were prepared and freshly frozen using isopentane chilled with liquid nitrogen and stored at −80 °C.

### Antibody labeling
**Preparation of PECAbs using one-step labeling.** For Alexa 488 azide (Thermo Fisher Scientific, A10266) and Alexa 594 azide (Thermo Fisher Scientific, A10270), antibodies were labeled with the fluorophores in a one-step reaction. To prepare NHS-SS-fluorophore couples, DBCO-SS-NHS ester (CONJU-PROBE, CP-2024) and the Alexa Fluor-azide were pre-reacted in DMSO for 3 h at room temperature. The fluorophore-SS-NHS ester products were stored at −20 °C until use. Purchased antibodies (100 μg) underwent buffer exchange to 1 × PBS using 10 K Amicon Ultra centrifugal filters (Sigma Aldrich, UFC501096) to remove sodium azide and glycerol. Antibodies were reacted with an equal molar quantity of the fluorophore-SS-NHS ester in 100 mM NaHCO$_3$/PBS for 1 h at room temperature. The products were purified using 10 K Amicon Ultra centrifugal filters (Sigma Aldrich) and stored in 50% glycerol/PBS at −20 °C until use.

**Preparation of PECAbs using two-step labeling.** For Janelia fluor 646 azide (Tocris Bioscience, 7088), antibodies were labeled with the fluorophore in two-step reactions. Antibodies, buffer-exchanged as above, were reacted with an equal molar quantity of DBCO-SS-NHS ester in 100 mM NaHCO$_3$/PBS for 1 h at room temperature. The products were purified using 10 K Amicon Ultra centrifugal filters (Sigma Aldrich) to remove unreacted DBCO-SS-NHS ester. The products were made up to 70 μL with PBS and then mixed with 0.7 μL of 10 mM fluor dye azide. After incubation for 24 h at 4 °C using a rotator, the products were purified using 10 K Amicon Ultra centrifugal filters (Sigma Aldrich) and stored in 50% glycerol/PBS at −20 °C until use.

**Preparation of PECAb-Fab.** Antibodies were reacted with DBCO-SS-NHS ester for 1 h at room temperature. The products were purified and reacted with azide-conjugated fluor dye. After incubation for 24 h at 4 °C using a rotator, the products were purified and stored in 50% glycerol/PBS at −20 °C until use.

**Preparation of fluor-labeled antibodies.** Antibodies were reacted with Alexa fluor 488 NHS ester (Thermo A20000) for 1 h at room temperature. The products were purified and stored in 50% glycerol/PBS at −20 °C until use.

**Preparation of Oligo labeled Fab.** The DBCO-SS-NHS ester conjugated Fabs described above were made up to 70 μL with PBS and then mixed with 350 μL of 10 μM of 5′ azide-modified oligonucleotide DNA 5′-ACGTTGAACGACACGTAAACGTTGAACGACACGTA-3′ containing 2 × readout probe binding unit across the AAA designed in the 'Secondary and readout probe design' section. After incubation for 24 h at 4 °C using a rotator, the products were purified using Microcon DNA Fast Flow (Millipore).

### Immunofluorescence
Cells or tissues were prepared on ibidi 96 well plates (ibidi 89626). The samples were fixed with 4% PFA (Electron Microscopy Science, 15710) for 10 min at room temperature. The fixed cells were washed three times with PBS and permeabilized with 0.5% TritonX100/PBS for 5 min at room temperature. Samples were blocked with Blocking One P (Nacalai Tesque) for 10 min at room temperature. Then, 2 μg/mL of primary antibodies diluted in 10% Blocking One/PBST were added to the sample and incubated for 45 min at room temperature. Samples were washed in PBST for 5 min 3 times and 1:1000 diluted secondary antibodies in 10% Blocking One/1 μg/mL Hoechst 33343/PBST were added. After incubation for 30 min, the samples were washed in PBST for 5 min 3 times before imaging. For Fig. 1d, contrast was adjusted as 200/2300(α-Tubulin, Conventional IF), 200/3200(α-Tubulin, PECAb), 200/6000(αTubulin, OligoDNA Ab), 120/800(pAKT, Conventional IF), 95/185(pAKT, PECAb), 100/1600 (pAKT, OligoDNA Ab), respectively.

### Erasing fluorophores
Fluorescent signals of samples stained with PECAbs were erased by incubation with 10 mM TCEP for 30 min. Another two erasing conditions, 3% H$_2$O$_2$/20 mM HCl in PBS for CycIF[6] or 3 M urea/3 M guanidinium chloride (GC)/70 mM TCEP in H$_2$O, pH adjusted to 2.5 with L-glycine for 4i[5] were used to compare sample preservation under the PECAb erasing conditions.

### Electron microscopy
Cells treated with antibody-stripping buffer 50 times were fixed with 2.5% glutaraldehyde in PBS overnight at 4 °C. After washing with PBS, the cells were post-fixed with 1% osmium tetroxide for 2 h. The cells were sequentially dehydrated in 50, 70, 90, 95, 99, and 100% EtOH, then incubated in propylene oxide, and then embedded in Epon resin (TABB, 3402). Ultrathin sections (80 nm) were stained with uranyl acetate for 5 min and with lead acetate for 10 min and then examined with a transmission electron microscope (FEI, TECNAI-20).

### Automatic device
The polymer coverslips, one side of which is covered with protective foil (ibidi GmbH, 10814), were modified to make an inner space and then the non-foil-protected side was attached to a 26 mm × 76 mm coverslip (ibidi GmbH, 10812) with double-sided adhesive tape. A 26 mm × 76 mm × 5 mm Borofloat33 (Matsunami) with punched 5 mm

diameter holes at the inlet and outlet of the microfluidic channel was attached on the foil protection side using double-sided adhesive tape and used as a microfluidics chip. The protective foil was then removed from the polymer glass, allowing to the retrieval of samples after seq-PECAb immunofluorescence for subsequent experiments. The inlet of the chip was connected with an AS-4150 Autosampler (JASCO) using PTFE tubes with a silicon tube as an adaptor. The microfluidics device was mounted on a DMi8 confocal microscope (Leica) equipped with a spinning disc confocal unit (Dragonfly200, ANDOR technology), a sCMOS camera (ANDOR technology, Zyla-4.2 plus), and a 63 × oil objective lens (Leica Plan Apo 1.40 NA). The timing of fluid delivery and imaging was controlled using custom Python scripts that interfaced with the autosampler and microscope operating software (Fusion software 2.3.0.44, ANDOR technology)

### Sequential Immunostaining (SeqIS) with PECAb

Biologically duplicated samples were prepared on a single microfluidics chip for all SeqIS and Seq-smFISH experiments.

**Preparation of cell line sample.** Cell culture inserts (ibidi, ib80209) were placed on silane-coated coverslips, and then cells were seeded to achieve 80–90% confluency. Following overnight incubation, the cells were washed with ice-cold 1% phosphatase inhibitor cocktail/PBS once and then fixed with freshly prepared 4% PFA (Electron Microscopy Science, 15710) for 10 min at room temperature. The fixed cells were washed three times with PBS and permeabilized with 0.5% TritonX100/PBS for 5 min at room temperature. The cells were treated with 1 µg/mL Hoechst33343 in Blocking One P (Nacalai Tesque, 05999-84) for 20 min at room temperature and placed on the microfluidics chip.

**Preparation of tissue sample.** Fresh frozen tissue was sliced at 10 µm thickness and placed onto a MAS-coated coverslip (Matsunami). The tissue slices were washed with ice-cold 1% phosphatase inhibitor cocktail/PBS once and fixed with freshly prepared 4% PFA containing 0.3% TritonX100 in PBS for 5 min at room temperature. The fixed tissues were washed three times with PBS and permeabilized with 0.3% TritonX100/PBS for 5 min at room temperature. The tissues were treated with 1 µg/ mL Hoechst33343 in Blocking One P (Nacalai Tesque) for 20 min at room temperature and placed on the microfluidics chip.

**Sequential immunostaining.** The microfluidics chip containing samples was first connected to the automated fluidics system. Then the region of interest (ROI) was registered using nuclei signals stained with Hoechst33343. Blank images were obtained before the first round of the staining cycle to determine the baseline background signals. Prior to sequential immunofluorescence, the staining order was carefully considered by the staining efficiency of the PECAbs. To assess the staining efficiency, IMR90 cells were stained with PECAbs, and scores were calculated by the laser power (%) × exposure time (ms) at which the fluorescence intensity of the specific signal was approximately 200. Then, SeqIS was performed by rearranging the PECAb staining order from highest to lowest. For SeqIS, the PECAb staining solutions were prepared for each cycle. Three different fluorescent-labeled PECAbs were selected from the Alexa488, Alexa594, and Janelia647 set and mixed together in a tube. Then, the storage buffer was replaced with PBS using Amicon Ultra centrifugal filters (Sigma Aldrich). Subsequently, Blocking One P (Nacalai Tesque) was added at 10% of the total volume, Hoechst 33343 was added at 1 µg/mL and used for the staining solution. PBST was used as the wash solution, and 10 mM TCEP with 0.2 µg/ml Hoechst 33343 in PBS was prepared as the erasing solution. These solutions were all pipetted into a 96-well plate and placed in the autosampler. The PECAb staining solution was incubated for 45 min for cell line samples and for 60 min for tissue samples at room temperature. After PECAb staining, the samples were washed with PBST for 5 min, three times to remove excess PECAbs and non-specific binding. Imaging was performed with a microscope (Leica DMi8) equipped with a spinning disc confocal unit (Dragonfly 200, ANDOR technology), a sCMOS camera (Zyla-4.2 plus, ANDOR technology), a 63 × oil objective lens (Leica Plan Apo 1.40 NA) or a 100 × oil objective lens (Leica Plan Apo 1.47 NA). Images were acquired by tiling a single slice in the x and y directions. After imaging, samples were incubated with erasing solution for 15 min, three times for cells or four times for tissue samples. After washing with PBST for 5 min, three times, imaging was performed with the same conditions as above to obtain subtraction images.

### Primary probe design and preparation for sequential RNA FISH

Primary probes were designed as reported[27] with some modifications. To screen the target sequences of primary probes, whole transcript sequences were obtained from the RefSeq database. For genes having multiple transcript isoforms, the longest isoform was selected as a representative gene. Using Bowtie (version 12.2.0) with option '-F28,1-v0', 28-mer substrings of each mRNA sequence were mapped to transcripts, and unique candidates were detected. The candidates with 45%–65% GC content were obtained. Genes with fewer than 24 candidates were filtered out. Within retained candidates, pairs with highly complementary sequences were detected by blastn in BLAST+ (version 2.2.31)[28] and excluded. Then 'word_size' was set to 10. Finally, genes having at least 24 candidates were obtained. Designed probes were ordered as oligo pools from Twist Bioscience. The oligo pools were amplified by limited PCR cycles using KOD One PCR Master Mix (TOYOBO, KMM-101) with the following primers, forward: 5′-CG GCAAACTACGCGAGCCATA-3′, reverse: 5′-AAGCTAATACGACTCACTA TAGG-3′. The amplified PCR products were purified using DNA Clean & Concentrator-5 (Zymo Research, D4014) according to the manufacturer's instructions. Then, in vitro transcription (IVT) was performed using the PCR products and MEGAshortscript™ T7 Transcription Kit (Thermo Fisher Scientific, AM1354). The transcribed RNAs were purified using MEGAclear™ Transcription Clean-Up (Thermo Fisher Scientific, AM1908). The purified RNA products were subjected to reverse transcription with Maxima H Minus Reverse Transcriptase (Thermo Fisher Scientific, EP0752), dNTPs (NEB, N0447L), and the forward primer containing a uracil nucleotide, 5′-CGGCAAACTACGCGAGCCA[U]A-3′. After reverse transcription, the products were treated with uracil-specific excision reagent (USER) enzyme (NEB, N5505S) to remove the forward primer. To purify single-stranded DNA (ssDNA) probes, the reverse transcription products were treated with RNaseA (Nacalai Tesque, 30100-31) for 0.5 h at 37 °C, then treated with ProK (Nacalai Tesque, 15679-06) for 2 h at 50 °C, followed by phenol-chloroform precipitation. The products were purified using DNA Clean & Concentrator-25 (Zymo Research, D4033). Then, the probes were dried in a centrifugal evaporator and resuspended in primary-probe hybridization buffer consisting of 40% formamide (Nacalai Tesque, 16229-95), 2 × SSC (Nacalai Tesque, 32146-91), and 10% dextran sulfate (Nacalai Tesque, 03879-14). The probes were stored at −20 °C until use.

### Secondary and readout probe design

To screen the secondary and readout probes, we first generated random sequences of 15 nucleotides. The candidates with 40%–60% of GC content and A or T at the 5′ end were obtained. Using web BLAST, candidates with high complementarity to the Human genomic plus transcript database were filtered out; the program was set to blastn, and models (XM/XP) and uncultured/environmental sample sequences were excluded. Using blastn in BLAST+, highly complementary pairs were excluded with the same option as the screening of primary probes. Finally, 243 sequences were obtained and employed as secondary or readout probes. Each secondary probe was designed to include sequences complementary to the primary probe, in addition to two identical readout probe binding sequences. The secondary probes

were synthesized and purified using an OPC column (Eurofins). Readout probes tagged with Alexa 488 and Alexa 647 at their 5′ ends were synthesized by Thermo Fisher Scientific, while those labeled with ATTO565 at the 5′ end was synthesized by Sigma Aldrich. All readout probes underwent purification by high-performance liquid chromatography (HPLC).

## Sample preparations for Seq-smFISH

Seq-smFISH samples were prepared based on the method of ref. 27 with some modifications. Sequential PECAb immunofluorescence-stained samples were permeabilized with 70% EtOH for 1 h. For tissue, samples were cleared with 8% SDS (Nacalai Tesque) in PBS for 30 min at room temperature. Then, primary probes were hybridized to the sample for 72 h at 37 °C in a humidified chamber. After primary probe hybridization, samples were washed with 40% formamide in 2 × SSC at 37 °C for 30 min. Samples were rinsed with 2 × SSC three times, then de-gassed 4% acrylamide (acrylamide: bis = 19:1, Kanto Chemical) in 2 × SSC hydrogel solution was added and incubated for 5 min at room temperature. The hydrogel solution was aspirated and 4% hydrogel solution containing 0.2% APS (Nacalai Tesque) and 0.2% TEMED (Nacalai Tesque) in 2 × SSC was added to the samples and then covered with a Gel-Slick (Lonza) treated coverslip. The samples were transferred into a homemade nitrogen gas chamber for 1 h. The hydrogel–hydrogel embedded samples were treated with protein digestion solution consisting of 50 mM Tris-HCl pH 8, 1 mM EDTA, 0.5% Triton-X100, 500 mM NaCl, 0.02 mg/mL ProK, 1% SDS in $H_2O$ for 3 h at 37 °C. After clearing, the samples were washed with 2 × SSC three times and then the nucleic acids in the samples were amine-modified using a Label IT Amine modifying kit (Mirus Bio) for 30 min at 37 °C. The samples were washed once with PBS and fixed by adding 4% PFA in PBS for 10 min at room temperature; then, 1 M Tris-HCl pH 8 was added for 10 min at room temperature to stop the fixation. The samples were washed with 4 × SSC and stored in 2 U/μL of SUPERase In RNase Inhibitor at 4 °C until imaging.

## Seq-smFISH imaging

The imaging platform used in the Seq-smFISH was as described above. Imaging was performed based on the method of ref. 27 with some modifications. The position of the sample was adjusted and registered to the same coordinates as the ROI captured in SeqIS imaging. Two microliters of 50 μM secondary probe and 1.5 μL of 100 μM corresponding readout probes were pre-incubated for 10 min at room temperature. Then 0.4 μL of the secondary-readout (S-RO) complex was diluted in 200 μL EC buffer [10% ethylene carbonate (Sigma E26258), 10% dextran sulfate (Sigma D4911), in 4 × SSC]. These complexes were prepared for the number of target RNAs to be detected. The S-RO hybridization solution, 4 × SSC, wash buffer (12.5% formamide, 0.1% TritonX100 in 2 × SSC), Hoechst buffer (0.1% TritonX100, 1 μg/mL Hoechst 33343 in 4 × SSC), stripping buffer (55% formamide, 0.1% TritonX100 in 2 × SSC), anti-bleaching buffer 50 mM Tris-HCl pH 8, 300 mM NaCl, 3 mM Trolox (Sigma 238813), 0.8% D-glucose (Nacalai Tesque 168060-25), 0.04 mg/mL catalase (Nacalai Tesque 07444-74) were all transferred into a 96-well plate and placed in the autosampler. The samples were first incubated with S-RO hybridization solution for 20 min at room temperature. After hybridization, samples were washed with 4 × SSC for 30 seconds, then wash buffer was added and incubated for 30 s to remove excess and non-specific binding probes. Then, the samples were washed once with 4 × SSC for 30 s, once with Hoechst buffer for 30 s, and then anti-bleaching buffer was added. Images were acquired at 1 μm thickness with 0.2 μm z-steps × 6 for the same ROI as the Seq-PECAb-IF imaging. After imaging, samples were incubated with stripping buffer for 1 min twice, washed with 4 × SSC for 30 s, and moved the next imaging cycle.

## Image analysis

**Image alignment.** SeqIS and Seq-smFISH images often exhibit misalignment because a several-micrometer misalignment occurs when the microscope stage returns to the original position after imaging. To quantify each cell in the same region of interest (ROI), image alignment was performed. Hoechst (nucleus staining) was added in the 4-channel imaging to align the produced images in each cycle. Because there is no misalignment within a single staining, image alignment was performed based on the Hoechst image. The alignment method employed a parallel translation alignment technique based on brightness similarity. Specifically, the images were shifted while calculating the similarity between images, and alignment was achieved by identifying the location where the similarity was highest. This process was repeated to achieve alignment for all consecutive immunostaining images.

**Nucleus region detection.** To obtain the regions for quantification, nucleus detection was performed on the Hoechst images. In this experiment, we utilized ellipse fitting to detect individual cell nuclei. Specifically, we employed a simplified version of the method proposed by Bise[29] to segment individual nucleus regions and applied ellipse fitting. The region segmentation was performed using a multi-thresholding method to extract candidate regions and represent their relationships in a tree structure. While using machine learning to compute a likelihood score and optimization to select the final segmentation results from candidate regions, in this experiment, we employed a rule-based selection method. Specifically, we selected the region with the maximum size among candidates that met predefined criteria for region size, maximum intensity within the region, and circularity. We used a tree structure to remove conflicting regions. The minimum region size was set to 1200 pixels, the maximum region size was set to 12,000 pixels, and the circularity threshold was set to 0.8. The brightness threshold was automatically determined based on the image. Ellipse fitting was performed using the *regionprops* function in MATLAB. For the obtained nucleus regions (N), we expanded the regions outward by 25 pixels in the major axis direction and 15 pixels in the minor axis direction to obtain the cytosol regions (C) outside the nuclei.

**Metric for SeqIS.** The metric was obtained for the nucleus cell i using the nucleus region $N_i$ and the cytosol region $C_i$ obtained in the previous step. For the aligned images, quantification metrics were computed for each channel within the same region. To eliminate background fluctuations, background images were acquired for each staining image, and these background images were subtracted from the staining images in each channel to create foreground images. The quantification metrics recorded included the average brightness values within the nucleus region $N_i$ and the cytosol region $C_i$.

*Metric for Seq-smFISH* Similar to the aforementioned SeqIS, the only difference lies in the metric calculation. Bright spots indicating single molecule RNA were detected with the same method as nucleus detection with different hyper-parameters. After extracting candidate regions, we detected bright spots as those satisfying the following conditions: within a predefined size range and having a maximum brightness value within the region exceeding a specified threshold. The minimum region size was set to 30 pixels, the maximum region size was set to 500 pixels, and the threshold for the maximum intensity was 0.2. For each cell nucleus, the numbers of bright spots in N and C were used as the metric.

**Metric for chromatin domain.** The Metric calculation method is different from SeqIS and Seq-smFISH. For the aligned images, the chromatin domains were detected and separated to 3 layered regions as S1, S2, and S3, then quantification metrics were computed for each channel within the same regions as described in '*Metric for SeqIS*'. The

quantification metrics recorded included the average brightness values within the S1, S2, and S3 regions.

## RNA sequencing

**scRNA-seq.** CEL-Seq2[30] was used for amplification and library preparation except that Maxima H minus reverse transcriptase (Thermo Fisher Scientific) was used for first-strand cDNA synthesis and Second Strand Synthesis Module (NEB) was used for double-strand cDNA synthesis. Individual propidium iodide negative cells were sorted into 384 wells. After 1st strand synthesis, all 384 wells were pooled and purified with two DCC-5 columns (Zymo Research). Libraries were sequenced on a NovaSeq6000 system (Illumina) and the following quantitative analysis was performed using 19 bp cell barcode reads (Read 1: NNNNNNNNNNCCCCCCCCC, N = UMI and C = Cell Barcode) and 81 bp insert reads (Read 2). Adaptor sequence and low-quality sequences of Read 2 were removed and reads of less than 20 bp were discarded using Trim Galore (ver.0.6.10). Insert reads were mapped to the GRCh38 reference using HISAT2 (ver. 2.2.1). Read counts for each gene were obtained by featureCounts (ver.2.0.4) and UMI duplications were removed by UMI-tools (ver. 1.1.4). Single-cell data visualization was performed using Seurat package (R version 4.2.1, Seurat_4.3.0.1). Wells with read counts lower than 100 or higher than 40,000, detected gene counts lower than 2000 and higher than 7000, and mitochondrial gene detection higher than 35% were filtered out. In total 1916 cells and protein-coding genes were used for further analysis. Read counts were log normalized and the most variable 2000 genes were identified. The data were scaled using ScaleData and linear dimensionality reduction (PCA) was performed. Non-linear dimensionality reduction (UMAP) was performed using PCA component of 1:20.

**RNA-seq.** BRB-seq[31] was performed for preparing libraries with some following modifications. Barcoded oligo-dT based primer(5′-G CCGGTAATACGACTCACTATAGGGAGTTCTACAGTCCGACGATCNNN NNNCCCCCCCCCTTTTTTTTTTTTTTTTTTTTTTTTTTTTTV -3′; (6)N = UMI, (9)C=cell barcode) was used for single-stranded synthesis and Second Strand Synthesis Module (NEB, #E6111) was used for double-stranded cDNA synthesis. In-house MEDS-B Tn5 transposase[32,33] was used for tagmentation and amplified by 10 cycles of PCR using Phusion High-Fidelity DNA Polymerase (Thermo Scientific, #M0530) and the following primers (5′- AATGATACGGCGACCACCGAGATCTAC ACindexGTTCAGAGTTCTACAGTCCGA-3′, 5′-CAAGCAGAAGACGGCA-TACGAGATindex GTCTCGTGGGCTCGGAGATGT-3′). 15 bp of barcode read (Read1) and 81 bp of insert read (Read2) were obtained on Illumina NovaSeq6000. Read1(barcode read) was extracted by using UMI-tools(ver.1.1.1) with the following command "umi_tools extract -I read1.fastq --read2-in=read2.fastq --bc-pattern=NNNNNNNNNNCCCC CCCCC --read2-stdout". Adaptor sequence and low-quality sequences were removed and read lengths below 20 bp were discarded by using Trim Galore (ver.0.6.7), and reads were mapped to the GRCm38 reference using HISAT2 (ver.2.2.1). Read counts for each gene and each sample were obtained by summarizing the bam file, which was produced by featureCounts(ver.2.0.1). Differentially expressed genes were extracted with DESeq2 (ver.1.34.0) using |log2FC| > 1 and padj <0.1 as threshold values.

## UMAP visualization and clustering

For protein expression data, quantified values were z transformed (mean 0 and s.d. 1) for each feature followed by arcsinh transformation. UMAP visualizations were performed using the *uwot* R package (version 0.1.10)[34] with parameter; metric = "cosine". Numbers of used PCs depended on the number of antibodies (features) of the dataset. In all cases, PC1 was considered to be the background signal and was therefore excluded from this analysis because it correlated highly with the total expression. In the same space as used for visualization,

clusters were determined by the Leiden algorithm[35] implemented in the *igraph* R package (version 1.5.0)[36].

For RNA expression data, the analysis was performed using *Seurat* (version 4.3.0)[37]. 'NormalizeData' was performed with 'scale.factor = 500'. Clusters were determined by the Leiden algorithm using 'FindClusters'.

## Pseudotime analysis

To infer pseudotime, PHATE embedding was performed using *phateR* (version 1.0.7)[38] with the parameter; 'mds.dist.method = "cosine"'. Based on the PHATE coordinates, pseudotime was calculated using *slingshot* (version 2.6.0)[39]. The clusters detected by k-means ($k = 5$) were used as input for 'slingshot' and 'start.clus' parameters of 'getLineage' were selected based on sample labels. The calculated pseudotime was scaled to [0,1].

## Silhouette analysis

For Fig. 3e, silhouette scores were calculated using PC2-20, and sample labels were used as cluster labels. For calculating silhouette scores of down-sampled datasets, PCA was performed against randomly sampled antibodies. The sampling was repeated 100 times.

For Supplementary Fig. 9d, silhouette scores were calculated using the coordinates of clustered cells in 2D space in the tissue. The labels were identical to those used for the clusters of the seq-smFISH expression space.

## Spatial correlation analysis

Spatial correlation coefficients of protein i and protein j were calculated as the weighted inner product of the two expression vectors:

$$SpatialCorCoef_{ij} = \mathbf{v}_i^T \mathbf{W} \mathbf{v}_j, \tag{1}$$

$$\mathbf{W} = \mathbf{A} / \sum_{k,l \in cells} \mathbf{A}_{kl}, \tag{2}$$

$$\mathbf{A}_{kl} = \begin{cases} \left(\frac{1}{d_{kl}}\right)^2 & (k \neq l) \\ 0 & (k = l) \end{cases}, \tag{3}$$

where $^T$ is the transpose operator and $\boldsymbol{v}_i$ is the vector of scaled expression levels of i-th protein in all cells, and $d_{kl}$ is the spatial distance between cell k and cell l. Definition of the weight matrix followed implementation of 'Seurat::RunMoransI'. To visualize the smoothed view of protein expression levels in the spatial coordinate, average expression levels of each $50 \times 50$ grid were calculated and smoothed by a $5 \times 5$ moving average filter three times.

## Statistics and reproducibility

Statistical analyses were performed using GraphPad Prism v8.4.3 software. One-way analysis of variance (ANOVA) followed by Dunnett's multiple comparisons test was used to compare multiple groups. $P < 0.05$ was considered statistically significant and $n$ is defined as biological replicates. Spatial omics experiments (SeqIS and Seq-smFISH) were performed with two biological replicates per condition. Other experiments were replicated as indicated in the legends. Image data from areas where samples were detached during SeqIS or Seq-smFISH experiments were excluded from the analysis.

## Reporting summary

Further information on research design is available in the Nature Portfolio Reporting Summary linked to this article.

# Data availability

The RNA-seq, scRNA-seq data generated in this study have been deposited in the GSE242887. The source and processed data in this

study have been deposited in the Zenodo repository [https://doi.org/10.5281/zenodo.10655938]. Materials availability was described in the Methods section. Antibodies used in this research are listed in Supplementary Data 1. The antibodies No. 185-210 were produced in the indicated studies[17,40–48]. Source data are provided with this paper.

## Code availability

All code for data analysis is available at https://github.com/tfwis/PECAb.

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

## Acknowledgements

We thank all members of the Ohkawa laboratory for helpful discussions and technical support. Computations were carried out using the computer resources offered under the category of Intensively Promoted Projects by the Research Institute for Information Technology at Kyushu University. We thank the research facilities of the Medical Institute of Bioregulation at Kyushu University for technical assistance. We thank Long Cai from California Institute of Technology for experimental support. This work was supported in part by the MEXT Cooperative Research Project Program, and CURE: JPMXP1323015486 for MIB, Kyushu University. This work was supported by JST FOREST JPMJFR2251 to K.To., JST CREST JPMJCR23N3 to H.Oc. and K.M., JST PRESTO JPMJPR2026 to K.M., Medical Research Center Initiative for High Depth Omics to Y.O., AMED BINDS JP23ama121020 to H.K. and JP22ama121017j0001 to Y.O., AMED JP22ama221513 to S.M., AMED JP23fk0210138 to K.To. and M.Nag., JSPS KAKENHI JP22H03538 and JP22K19314 to K.To., JP22J20655 to T.F., JP21H05292 and JP23H02394 to A.H., JP22H04696, and JP23H04288 to K.M., JP22H02609, JP22H04694, JP23H04286, and JP24H02326 to H.Oc., JP20H05609 to H.K., JP18H05528 to H.K. and Y.O., JP23H00372, JP22H04676, JP22K19275, JP23H00372 and JP24H02323 to Y.O., JP22K15084 to H.Oh., Cancer Research UK (C9545/A29580) and BBSRC (BB/S013466/1, BB/T013486/1) to M.Nar.

## Author contributions

Y.O., K.To. and F.T. conceived the project. K.To., K.H., Y.T., H.Oc., H.Oh., K.A., R.M., K.Ta. and A.H. designed and performed experiments, and analyzed the data. T.F., R.B., K.M. and S.U. analyzed the imaging data. T.T., S.M., M.Nag., H.K. and M.Nar. provided materials. Y.O., K.To. and A.H. wrote the manuscript with input from all authors.

## Competing interests

The authors declare no competing financial and non-financial interests except for K.To. and Y.O., who are involved in a patent related to PECAb that is now pending (Japan Patent Application 2020-199800).
