## [Peer Review File · Nature Communications]

Reviewers' Comments:

Reviewer #1:

Remarks to the Author:

[Brief Paper Summary]

Spatial -omics has been a rapidly growing field, which promises to elucidate cell-cell communication and interactions. While there have been many advances in sub-cellular resolution imaging of transcripts, the technology for identifying precise protein locations has been subject to technical limitations; Multiplexed immunofluorescent imaging allows for the acquisition of hundreds of protein locations at once, but faces difficulties with target specificity. To fill this research gap, Tomimatsu et al. have developed a novel method of antibody imaging that has high throughput while maintaining high target specificity, and can be combined with FISH transcriptomic methods to obtain a comprehensive picture of cell processes.

Experiments were conducted to establish baseline validity of SeqIS as a method, confirming that PECAb target specificity was comparable to non-multiplexed immunofluorescent methods, that accumulated PECABs after many imaging cycles did not hinder protein detection, and that results were repeatable across replicates. With this framework established, further experiments were conducted to demonstrate the ability of SeqIS data to provide meaningful data pertaining to cell states. This was accomplished by triggering a cell state change in a population, either by inducing senescence or by inducing an epithelial-to-mesenchymal transition, measuring 206 or 27 protein targets, respectively, and then quantifying the changes in these cell states across pseudotime. To better apply this technology in applications where there is not an existing body of well-characterized antibodies, Tomimatsu et al. designed a combined SeqIS – seq-smFISH experiment, first validating that the necessary chemical washes would not interfere with transcript detection and establishing a method to perform protein-clearing. This technique was then scaled up and applied to a uterine carcinosarcoma sample, where cell-state clusters were identified with seq-smFISH and cell-types identified with SeqIS.

[Major Edits]

There are a number of places where sub-figures could be better integrated into the narrative to strengthen the argument being made. The plots in the bottom half of 1D are addressed in neither the figure caption nor the text; It would also be beneficial to mention that the plots are derived from the full antibody set (ie the rightmost image) in the caption. Additionally, while the images at the top of the sub-figure clearly illustrate the presence of fluorescent signal in the nuclei with the oligo-based probes, the intent of the plots is not as clear. It would be beneficial to add a sentence to the results text clarifying something to the effect of 'the distribution of high-intensity pixels from PECAb staining more closely resemble that of Conventional IF than the low-specificity Oligo-DNA Antibody staining'.

Similarly, it would be constructive to either explain the role of the PHATE plots in figures 3F and 4G in the results text or to remove the figures. The results text does a good job justifying the use of pseudotime by highlighting ways that known cell markers line up with the plotted pseudotime trajectory, but no discussion is given to the PHATE embedding or its significance.

In the initial experiment where smFISH was integrated to seqIS (the second half of "Expansion of Analysis through Integration of PECAb with Seq-smFISH"), what was the rationale for the choice of target transcripts? While the following experiment describes the rationale behind target selection and the methods describe probe design, there is no provided rationale for these probes. Are there any localization patterns we would expect to see with these transcripts, or specific cell states where they would have altered abundances?

smFISH methods can be prone to high false-positive rates due to the fact that any high-intensity pixels are treated as signal and, unlike barcoded FISH methods, there is no way to perform error-correction or quantify the false positive rate. While it is detailed in Figure 2D-2 that background subtraction is performed for SeqIS image processing, is an analogous procedure followed for the smFISH images? Alternatively, have there been any analyses done to quantify average background

signal compared to high-intensity pixels, and how these metrics change when image processing measures (such as clipping and scaling values, applying a high pass Gaussian filter, or Richardson-Lucy deconvolution) are applied?

Arguments made on the basis of populations of cells clustering together, such as in Figure 6F, would be made stronger if clustering was quantified in some way. One such method to do this would be to find Ripley's K Statistic for all radii $0 < r < (\text{total pixels in image} / 2)^{0.5}$; A population that is clustered at radius r will have a higher score than a random Poisson process.

[Minor Edits]

While it is appreciated that each figure number corresponds to one experiment, the resultant figures have many subheadings and compress a lot of information into a small print space. As a result, many figures suffer from readability issues due to low-resolution images and small font sizes. For example:

- Figure 1A is not readable, all atom labels should be 6pt at the final print size.
- Figure 4B is too small to identify colors in the UMAP plot and font sizes are too small.
- Figure 6J, bottom half image is too low-resolution to identify unique colors per EMT marker.
- There are multiple tables used to illustrate selected antibodies and their expected localization (1E, 2B, 3B, 4C, 6D, 6H); They are titled inconsistently and are too small to read.
- Most heatmaps (4B right, 4E, 6I) need larger fonts and/or condensed labeling (such as in 3D).

Resolving these resolution issues may require splitting each figure into multiple figures and/or moving select sub-figures to the supplement.

Manuscript line 300 cites "Figure 5I", which should be "Figure 5G".

Reviewer #2:

Remarks to the Author:

This paper by Tomimatsu, Fujii and colleagues developed a sophisticated method termed Precise Emission Canceling Antibodies (PECABs) for highly multiplexed imaging capturing specific cell state, which enable high-specificity sequential imaging using hundreds of antibodies, allowing for reconstruction of the spatiotemporal dynamics of signaling pathways. In addition, by combining with seq-smFISH technique, this method can effectively classify cells and identify their signal activation states in human tissues. Together, the newly developed technique provides a comprehensive platform for analyzing complex cell processes.

Overall, this is an impressive series of experiments in several regards. The experiments are technically sophisticated. In particular, the convergence across techniques added a large degree of rigor to these findings, and increase confidence in their conclusion. There are a considerable number of experiments, and the investigation of this novel technique is extremely thorough and useful for the cell biology field. I do not have any other substantive concerns, and recommend this paper be accepted with minor revisions. A trivial criticism, yet one that might improve readability, is to greatly standardize use of acronyms throughout the text (eg, DBCO-SS-NHS showing in Line 99 on Page 4 ; TCEP showing in Line 275 on Page 9 etc.).

-The authors should provide the detailed statistical analysis methods in the "Materials and Methods" part.

-In this study, the authors used the uterine carcinosarcoma tissue to test the reliability of their novel technique together with seq-smFISH method. But they did not introduce the detailed information about tissue preparation before PECABs detection. For example, the carcinosarcoma tissue used for the experiment is fresh frozen slices or paraffin slices? Whether both samples are suitable for the PECABs technique?

-The anti-bleaching buffer is pretty tricky in this study, did the authors compare the results using

different conditions of anti-bleaching buffer with different concentrations of reagents?

Reviewer #3:

Remarks to the Author:

This paper proposed PECAb, cleavable fluorescent-dye-labeled antibodies with a mild cleaving condition, for highly multiplexed cyclic imaging. The authors successfully demonstrated various multiplexed fluorescent imaging within a single specimen and validated the PECAb in terms of multiplexing capability, cell loss, staining quality, and steric hindrance, compared with other existing multiplexing strategies. As far as I know, this is a demonstration of the largest instance of protein multiplexing ever reported. I truly admire the efforts and dedication of the authors; however, the manuscript lacks quantitative studies on the staining patterns of the PECAb antibodies and details on how the background signals or non-specific binding signals accumulate during repeated staining.

1. Regarding antibody specificity, in the abstract, the authors state that antibody specificities are compromised in other multiplexed imaging techniques, whereas their technique does not have this effect. However, overall, the manuscript lacks two things regarding this point: details on the antibody staining pattern (or specificity or degree of non-specific binding) of their antibodies, and antigen degradation in repeated applications of TCEP. The authors provided a qualitative analysis on the specificity of their antibodies, but most of the validation work the authors addressed was based on highly specific antibodies, such as tubulin or TOM20. I do not think that PECAB needs to work for all antibodies, but providing a quantitative measure of the staining pattern or specificity would be highly beneficial to potential readers who design experiments using this technique. One option would be staining the same specimen with regular indirect staining and PECAB simultaneously, using different colors.

2. Related to the epitope damage, the authors compared the staining patterns of different cells in Figure S4D. I did not find it clear how this experiment was performed. Was this experiment performed using the same specimen or separate specimens? If performed using the same specimen, this means that a single specimen could be stained with the same antibodies in two different rounds, such as Lamin A in the 1st and 56th cycles. Then, showing the images of the same cells in these two cycles, not different cells, would be a more accurate way to show the epitope damage. If this experiment was performed in different specimens, as the same proteins cannot be stained multiple times, any quantitative measurements made over multiple cells would be needed.

3. Regarding background signals or non-specific binding, Figure 1D shows lower nuclear staining of PECAb, compared with oligo DNA, but the results of the application of PECAb-Fab showed a higher level of autofluorescence (or non-specific binding?), when compared with the results of the application of the regular secondary antibody alone. Does this result show a higher level of non-specific binding of PECAb?

4. In Line 160, the authors claim that the accumulation of background signals was not observed across 21 cycles of staining, but in Line 185, they say that the background signals were accumulated beyond 20 cycles. Fig S4C shows that background signals were gradually accumulated from the initial round. Line 285, again, states that there is a background caused by protein signals. Please clarify the origin of the background and their levels. As only primary antibodies, which generally show lower signal intensities than indirect staining, were used in this study, the increase in the background signals could be a problem in later rounds.

5. Fluorophore-conjugated primary antibodies were used in this study. Considering that more than 200 primary antibodies were used, the conjugation procedures may be highly different among them, as the concentrations of commercial primary antibodies are different, as are their buffers (e.g., buffers w/ or w/o stabilizer proteins). However, the methods section on the conjugation of primary antibodies and fluorophores is very simple. Did the authors perform a conjugation optimization for each antibody, such as buffer exchange or degree of labeling optimization? If so, providing such information would be highly beneficial to potential readers who want to use this

method.

6. In their discussion, the authors highlighted the possibility of steric hindrance occurring among antibodies, and they took meticulous steps to optimize the staining order. Specifically, they prioritized the application of antibodies with lower affinities. It would be helpful for prospective users of this technique if the authors could provide details on how they assessed antibody affinity.

7. Based on the findings presented in Figure S6A, it is evident that washing with PBS results in an approximately 50% cell loss, while the use of TCEP treatment hardly causes any cell loss. It is not easy to understand how TCEP treatment induces less cell loss than a simple PBS wash.

8. Line 449 provides multiple erasure solutions. Which solution was used for signal removal in this paper? What are the main differences among these erasing solutions, in terms of erasing performance, bio-sample integrity, and sample damage?

REVIEWER COMMENTS

We express our deepest gratitude to the three reviewers who reviewed the manuscript. Their comments were full of suggestions to improve the manuscript, and we have responded to them in consultation with each of the authors. Each comment is addressed below in order of importance. The reviewers' comments are in blue and our responses are in black.

Reviewer #1 (Remarks to the Author):

[Major Edits]

There are a number of places where sub-figures could be better integrated into the narrative to strengthen the argument being made. The plots in the bottom half of 1D are addressed in neither the figure caption nor the text; It would also be beneficial to mention that the plots are derived from the full antibody set (ie the rightmost image) in the caption.

We apologize for the omission in the caption; we have noted in the figure caption that the plots in the lower half of Fig. 1D were obtained from the complete antibody set on the far right, and we have also added the following to the text.

Figure caption (Line 1053 on Page 32)

D. Top panels, immunofluorescence images of IMR90 cells stained with indicated antibodies. Scale bars, 10 μ m. Bottom panels, the profile of fluorescence intensity on the yellow line drawn on the immunostaining images using an indicated full antibody set.

Text (Line 122 on Page 4)

We then determined the fluorescence intensity of line profiles drawn across the immunostaining images using images stained with the full antibody set.

Additionally, while the images at the top of the sub-figure clearly illustrate the presence of fluorescent signal in the nuclei with the oligo-based probes, the intent of the plots is not as clear. It would be beneficial to add a sentence to the results text clarifying something to the effect of 'the distribution of high-intensity pixels from PECAb staining more closely resemble that of Conventional IF than the low-specificity Oligo-DNA Antibody staining'.

To clarify the intent of the figure, we have specified in the text that the presence of fluorescent signals in the nuclei with the oligo-based probes and the distribution of high-intensity pixels with PECAb staining are closer to conventional immunofluorescence than for the low specificity oligo-DNA antibody staining. We have added the following to the text.

Text (Line 123 on Page 4)

These line profiles indicated that the distribution of fluorescence intensity from PECAb staining more closely resembled that of conventional IF than the oligo-DNA antibody staining (Fig. 1D bottom).

Similarly, it would be constructive to either explain the role of the PHATE plots in figures 3F and 4G in the results text or to remove the figures. The results text does a good job justifying the use of pseudotime by highlighting ways that known cell markers line up with the plotted pseudotime trajectory, but no discussion is given to the PHATE embedding or its significance.

We apologize for the lack of explanation. As the reviewer points out, we did not specify the role of the PHATE plot. Therefore, we have followed your suggestion and added the following explanation to the Results.

Text (Line 223 on Page 7).

To evaluate the dynamics of cell state changes, the quantified IMR90 cell dataset for senescence days 0, 3, and 6 was embedded using PHATE, and the pseudotime was constructed using PHATE coordinates (Fig. 3F). We found a correlation between the time samples were collected and pseudotime progression (Fig. 3G)

In the initial experiment where smFISH was integrated to seqIS (the second half of “Expansion of Analysis through Integration of PECAb with Seq-smFISH”), what was the rationale for the choice of target transcripts? While the following experiment describes the rationale behind target selection and the methods describe probe design, there is no provided rationale for these probes. Are there any localization patterns we would expect to see with these transcripts, or specific cell states where they would have altered abundances?

We apologize for the lack of explanation. In this experiment, transcripts expressed at high, low and medium levels were selected from the RNAseq data to evaluate the dependence of transcripts on expression level. We have added Fig. S8C and text to explain the gene selection in the results. Changes

in localization patterns and abundance of these transcripts were not included in this evaluation. Although not included in this evaluation, we believe that it would be useful to include genes that change localization of expression in future analyses. Therefore, we have included this point in the Discussion section.

Fig. S8C

Caption (Extended data, Line 82 on Page 9)

C. Selected genes for seq-smFISH in Fig.5. Genes expressed at high, medium, and low levels were selected from IMR90 cell RNA-seq data (GSE72404), indicated by red dots.

Text (Line 310 on Page 10)

Using a public RNA-seq dataset for IMR90 cells (20), primary probes were designed to detect randomly selected genes expressed at high, medium, and low levels (Fig. S8C).

Text in Discussion section (Line 402 on Page 12)

We have achieved a spatial multi-omics approach combining PECAbs and RNA Seq-smFISH, enabling the simultaneous acquisition of protein and RNA expression datasets with spatial information. Transcribed RNAs are not only translated to proteins but also contribute to the regulation of gene expression by interacting with proteins. PECABs can detect protein localization with high specificity and, therefore, have potential application in the analysis of RNA-protein complex dynamics during cell state changes.

smFISH methods can be prone to high false-positive rates due to the fact that any high-intensity pixels are treated as signal and, unlike barcoded FISH methods, there is no way to perform error-correction or quantify the false positive rate. While it is detailed in Figure 2D-2 that background subtraction is performed for SeqIS image processing, is an analogous procedure followed for the smFISH images? Alternatively, have there been any analyses done to quantify average background signal compared to high-intensity pixels, and how these metrics change when image processing measures (such as clipping and scaling values, applying a high pass Gaussian filter, or Richardson-Lucy deconvolution) are applied?

Your point is very important. Our seq-smFISH method differs from the commonly used smFISH protocol in that it allows background to be minimized and, consequently, it does not require background subtraction in image processing (see Fig. S8D unstained). We have added related data to Fig. S8D and amended the caption. Specifically, protein removal was carried out during sample preparation to remove the background source. Details of the procedure are given in the 'Sample preparations for Seq-smFISH' subsection in the Materials and Methods. We also added related text to the Results section (Line 306 and Line 316).

For smRNA spot detection, we applied a Gaussian filter (variance 1.5, kernel size 21×21) to the input image, and normalized the pixel intensities to a range of 0 to 1. This process effectively suppresses the effect of noise. Furthermore, our multi-threshold approach detects many candidate regions and determines the final bright point regions based on the size of the detected areas. This allows for the robust acquisition of bright points, providing resilience against noise.

Fig. S8D

Caption (Extended data, Line 84 on Page 9)

D. Detection of specific RNA spots. Dotted square regions are magnified in the bottom panels. Red crosses indicate called RNA spots.

Text (Line 306 on Page 9)

Protein clearing was then performed after the sample was embedded in the hydrogel.

Text (Line 316 on Page 10)

For quantification of Seq-smFISH, although background noise was minimized by protein clearing (Fig. S8D, unstained), the ReadOut probe signal can be seen to be different from that of RNA spots (Fig. S8D, center). Therefore, the RNA spots were determined by adjusting the brightness and the maximum and minimum pixel size parameters to distinguish noise from RNA spots (Fig. S8D right).

Arguments made on the basis of populations of cells clustering together, such as in Figure 6F, would be made stronger if clustering was quantified in some way. One such method to do this would be to find Ripley's K Statistic for all radii $0 < (total\ pixels\ in\ image / 2)^{0.5}$; A population that is clustered at radius r will have a higher score than a random Poisson process.

Thank you for your advice. We have performed an additional evaluation of clustering using silhouette analysis. The results are shown in Fig. S9D: a score > 0 for the Silhouette coefficient indicates proximity to cells in the same cluster, and a score < 0 indicates proximity to cells in other clusters. Clusters 1 and 2 are more prone to aggregating cells with similar profiles in the tissue compared with Clusters 3, 4, and 5. This result indicated that the method is valid for analyzing the spatial distribution of cells in the tissue. To emphasize these results, we have added Fig. S9D and the following sentences to the Results and Materials and Methods sections.

Text (Line 344 on Page 10)

According to silhouette analysis, Clusters 1 and 2 were more prone to aggregating cells with similar profiles in the tissue compared with Clusters 3, 4, and 5 (Fig. S9D).

Text (Line 797 on Page 23)

For Figure S9D, silhouette scores were calculated using the coordinates of clustered cells in 2D space in the tissue. The labels were identical to those used for the clusters of the seq-smFISH expression space.

Fig. S9D

Caption (Extended data Line 91 on Page 10)

Fig. S9D. Silhouette coefficient of clustered cells to analyze spatial occupancy. A score > 0 indicates proximity to cells in the same cluster, and a score < 0 indicates proximity to cells in other clusters.

[Minor Edits]

While it is appreciated that each figure number corresponds to one experiment, the resultant figures have many subheadings and compress a lot of information into a small print space. As a result, many figures suffer from readability issues due to low-resolution images and small font sizes. For example:

- Figure 1A is not readable, all atom labels should be 6pt at the final print size.

We have updated the font size to 6 pt.

- Figure 4B is too small to identify colors in the UMAP plot and font sizes are too small.

We have updated the dot size in the UMAP, and font size to 6 pt.

- Figure 6J, bottom half image is too low-resolution to identify unique colors per EMT marker.

We have replaced the image with one of higher resolution.

- There are multiple tables used to illustrate selected antibodies and their expected localization (1E, 2B, 3B, 4C, 6D, 6H); They are titled inconsistently and are too small to read.

We have updated all table titles to a font size of 6 pt.

- Most heatmaps (4B right, 4E, 6I) need larger fonts and/or condensed labeling (such as in 3D).

We have updated the font size to 6 pt.

Resolving these resolution issues may require splitting each figure into multiple figures and/or moving select sub-figures to the supplement.

Manuscript line 300 cites “Figure 5I”, which should be “Figure 5G”.

We have corrected the error. Thank you.

Reviewer #2 (Remarks to the Author):

This paper by Tomimatsu, Fujii and colleagues developed a sophisticated method termed Precise Emission Canceling Antibodies (PECABs) for highly multiplexed imaging capturing specific cell state, which enable high-specificity sequential imaging using hundreds of antibodies, allowing for reconstruction of the spatiotemporal dynamics of signaling pathways. In addition, by combining with seq-smFISH technique, this method can effectively classify cells and identify their signal activation states in huma tissues. Together, the newly developed technique provides a comprehensive platform for analyzing complex cell processes.

Overall, this is an impressive series of experiments in several regards. The experiments are technically sophisticated. In particular, the convergence across techniques added a large degree of rigor to these findings, and increase confidence in their conclusion. There are a considerable number of experiments, and the investigation of this novel technique is extremely thorough and useful for the cell biology field. I do not have any other substantive concerns, and recommend this paper be accepted with minor revisions.

We appreciate your positive comments on the paper and your suggestions on how to improve the manuscript. Our responses to your specific comments are below. We apologize for the inconsistency of abbreviations, which we have now standardized.

A trivial criticism, yet one that might improve readability, is to greatly standardize use of acronyms throughout the text (eg, DBCO-SS-NHS showing in Line 99 on Page 4 ; TCEP showing in Line 275 on Page 9 etc.).

We have standardized use of acronyms throughout the text, as suggested.

The authors should provide the detailed statistical analysis methods in the “Materials and Methods” part.

We apologize for the insufficient description of the statistical analysis. We have added the following description (Line 815 on Page 24).

Text (Line 815 on Page 24)

‘Statistics and reproducibility

Statistical analyses were performed using GraphPad Prism v8.4.3 software. One-way analysis of variance (ANOVA) followed by Dunnett’s multiple comparisons test was used to compare multiple groups. $P < 0.05$ was considered statistically significant and n is defined as biological replicates. Spatial omics experiments (SeqIS and Seq-smFISH) were performed with two biological replicates per condition. Other experiments were replicated as indicated in the legends. Image data from areas where samples were detached during SeqIS or Seq-smFISH experiments were excluded from the analysis.

In this study, the authors used the uterine carcinosarcoma tissue to test the reliability of their novel technique together with seq-smFISH method. But they did not introduce the detailed information about tissue preparation before PECABs detection. For example, the carcinosarcoma tissue used for the experiment is fresh frozen slices or paraffin slices? Whether both samples are suitable for the PECABs technique?

We apologize for the lack of detailed information on tissue preparation before PECAB detection. We have added the following text to the Materials and Methods section.

Text (Line 539 on Page 16)

Preparation of tissue sample. Fresh frozen tissue was sliced at 10 μm thickness and placed onto a MAS-coated coverslip (Matsunami).

The preparation of fresh-frozen tissue is described in the **Human Tissues** subsection (Line 428 on Page 13) and **Mouse Tissues** subsection (Line 436 on Page 13) subsections of the Materials and Methods.

The anti-bleaching buffer is pretty tricky in this study, did the authors compared the results using different conditions of anti-bleaching buffer with different concentrations of reagents?

As you pointed out, we understand that the buffer we used in this study is more complex than the one used in single-molecule FISH. We used the same anti-bleaching buffer used in the SeqFISH+ method described by Eng et al. (Nature 2019) because this buffer allowed us to detect the stability of single-

molecule RNA. Therefore, we did not need to optimize the condition. We have added the following sentence to the Materials and Methods section, indicating that we performed imaging based on the method of Eng et al. (Nature 2019), so that readers can refer to the reference of this method.

Text (Line 651 on Page 19)

Imaging was performed as previously described (27) with some modifications.

Reviewer #3 (Remarks to the Author):

This paper proposed PECAb, cleavable fluorescent-dye-labeled antibodies with a mild cleaving condition, for highly multiplexed cyclic imaging. The authors successfully demonstrated various multiplexed fluorescent imaging within a single specimen and validated the PECAb in terms of multiplexing capability, cell loss, staining quality, and steric hindrance, compared with other existing multiplexing strategies. As far as I know, this is a demonstration of the largest instance of protein multiplexing ever reported. I truly admire the efforts and dedication of the authors; however, the manuscript lacks quantitative studies on the staining patterns of the PECAb antibodies and details on how the background signals or non-specific binding signals accumulate during repeated staining.

Thank you very much for your kind comments. We found your comments very helpful in enabling us to improve the clarity and readability of our paper. Our responses to each comment are below.

1. Regarding antibody specificity, in the abstract, the authors state that antibody specificities are compromised in other multiplexed imaging techniques, whereas their technique does not have this effect. However, overall, the manuscript lacks two things regarding this point: details on the antibody staining pattern (or specificity or degree of non-specific binding) of their antibodies, and antigen degradation in repeated applications of TCEP. The authors provided a qualitative analysis on the specificity of their antibodies, but most of the validation work the authors addressed was based on highly specific antibodies, such as tubulin or TOM20. I do not think that PECAB needs to work for all antibodies, but providing a quantitative measure of the staining pattern or specificity would be highly beneficial to potential readers who design experiments using this technique. One option would be staining the same specimen with regular indirect staining and PECAB simultaneously, using different colors.

We agree to show as many staining patterns and specificities of PECABs as possible for the benefit of the reader. Therefore, we have added staining images of an additional 100 antibodies in Fig. S1 and related text to the Results (Line 127 on Page 4). We believe that the presentation of these data sets

improves the reliability of our data and supports our presentation. We thank you for suggesting the inclusion of these data.

Text (Line 127 on Page 4)

For other examples, 100 specific PECAb-stained images are shown in Fig. S1.

Fig. S1

Caption (Extended data Line 17 on Page 2)

Fig. S1 Evaluation of PECAbs. Samples (IMR90, A549, and mouse TA tissue) were stained with PECAbs as the 1st Ab and captured using 2nd Abs conjugated with fluorophores. Different

combinations of fluo-dyes were used for PECAbs and 2nd Abs. Scale bars = 50 μ m.

2. Related to the epitope damage, the authors compared the staining patterns of different cells in Figure S4D. I did not find it clear how this experiment was performed. Was this experiment performed using the same specimen or separate specimens? If performed using the same specimen, this means that a single specimen could be stained with the same antibodies in two different rounds, such as Lamin A in the 1st and 56th cycles. Then, showing the images of the same cells in these two cycles, not different cells, would be a more accurate way to show the epitope damage. If this experiment was performed in different specimens, as the same proteins cannot be stained multiple times, any quantitative measurements made over multiple cells would be needed.

We apologize for the lack of explanation for Fig. S4D. The experiment was performed on separate specimens because it focused on whether the same staining pattern can be obtained in the 1st and later cycles, even if the sample is covered by accumulated antibodies. To clarify, the experiment was performed on different specimens. We have added relevant text to the Fig. S5 caption.

In response to the reviewer's concern, we performed additional experiments to evaluate the relevance of epitope damage to antibody staining patterns. Separate IMR90 cell specimens were prepared and treated with 10 mM TCEP 50 times and then stained with indirect immunofluorescence. Then cells were segmented and the fluorescence intensities in the segmented areas were quantified (see Fig. S5E). The effect of epitope damage depended on the antibody. For example, fluorescence intensity and specificity were not affected for the high-quality α -Tubulin monoclonal antibody in TCEP-treated samples. In contrast, staining with an anti-Lamin A polyclonal antibody, which is a mixture of antibodies of different quality, resulted in lower fluorescence intensity in TCEP-treated samples. We therefore used monoclonal antibodies as most of the PECAbs. We believe this provides evidence to reinforce the feasibility of the procedure with regard to staining concerns caused by epitope damage. We have added relevant text to the Results section (**Line 197 on Page 6**) and the Fig. S 5E caption.

Caption (Extended data Line 49 on Page 6)

D. Comparison of PECAb staining patterns. The 1st cycle and indicated latter cycle images were obtained from separate specimens.

Text (Line 197 on Page 6)

Additionally, to evaluate the relevance of epitope damage to antibody staining patterns, IMR90 cells

were treated with 10 mM TCEP 50 times and stained with anti-LMNA (polyclonal) and anti- α -tubulin (monoclonal) antibodies. Cells were segmented and the fluorescence intensity (FI) was quantified. Although the FI signal using monoclonal anti- α -tubulin was not decreased, signal using polyclonal anti-Lamin A, which is a mixture of antibodies of different quality, was decreased by TCEP treatment (Fig. S5E). These results indicate that the effect of epitope damage on staining depends on the quality of the antibody. Monoclonal antibodies were used to prepare most PECABs.

Fig. S5E

Caption (Extended data Line 52 on Page 6)

E. Top panel: fixed IMR90 cells were incubated with indicated 1st antibodies and then captured with specific 2nd antibodies. Bottom panel: fluorescence intensity of staining was measured. n = 7468, 3875, 3036, 2611, 2824 cells from two biological replicates of No 1st Ab, Untreated, PBS \times 50 times, 10 mM TCEP \times 50 times, and 100 mM TCEP \times 50 times, respectively.

3. Regarding background signals or non-specific binding, Figure 1D shows lower nuclear staining of PECAb, compared with oligo DNA, but the results of the application of PECAb-Fab showed a higher level of autofluorescence (or non-specific binding?), when compared with the results of the application of the regular secondary antibody alone. Does this result show a higher level of non-specific binding of PECAb?

We apologize for any misinterpretation due to inadequate explanation of the figures presented. We adjusted the contrast for each antibody to emphasize specificity but not sensitivity. To evaluate non-specific binding of PECAb as you mentioned, the contrast should have been consistent. Therefore, here we show the same contrast in the lower left for images obtained under the same imaging conditions. The non-specific signal of the PECAb Fab-treated sample is detected at a lower level, similar to conventional IF. Although this result indicates that staining with PECAb results in immunostained images with less non-specific noise, to avoid misleading, we added the following individual contrast values for Fig. 1D in the Materials and Methods section.

Text (Line 491 on Page 15)

For Fig. 1D, the contrast was adjusted to 200/2300(α Tubulin, Conventional IF), 200/3200(α Tubulin, PECAb), 200/6000(α Tubulin, OligoDNA Ab), 120/800(pAKT, Conventional IF), 95/185(pAKT, PECAb), 100/1600 (pAKT, OligoDNA Ab), respectively.

4. In Line 160, the authors claim that the accumulation of background signals was not observed across 21 cycles of staining, but in Line 185, they say that the background signals were accumulated beyond 20 cycles. Fig S4C shows that background signals were gradually accumulated from the initial round. Line 285, again, states that there is a background caused by protein signals. Please clarify the origin of the background and their levels. As only primary antibodies, which generally show lower signal intensities than indirect staining, were used in this study, the increase in the background signals could

be a problem in later rounds.

As you point out, background accumulates in the later cycles. This occurs because the erasing efficiency of the PECAb signal is not 100%. Faded antibody specific-signals are observed in TCEP-treated samples by imaging with long exposures. Accumulation of these signals due to SeqIS results in the background increasing with the number of cycles.

For the 21 cycles of staining shown in Fig. 2, the text states that ‘Accumulation of background was not observed across the 21 cycles of staining’, but there is in fact a trace accumulation of background. We have corrected the text as follows:

Text (Line 166 on Page 6)

Accumulation of slight backgrounds was observed across 21 cycles of staining.

Also, related to Reviewer 3 remark 6, we have tried to minimize the effect of this background for quantification by the following methods: (i) subtracting the erasing image from the stained image, and (ii) using brighter antibodies for the latter cycle. Point (i) was already addressed in the Discussion section (p12 line 1). For point (ii), we added the following text to the Materials and Methods section (p16 line 33).

Text (Line 551 on Page 16)

Prior to sequential immunofluorescence, the staining order was carefully considered by the staining efficiency of the PECAbs. To assess the staining efficiency, IMR90 cells were stained with PECAbs, and scores were calculated by the laser power (%) \times exposure time (ms) at which the fluorescence intensity of the specific signal was approximately 200. Then, SeqIS was performed by rearranging the PECAb staining order from highest to lowest.

For line 285 (Line 301 on Page 9), we would like to inform to reviewer that the protein removal is required to detect single molecule RNA spots regardless of the SeqIS.

5. Fluorophore-conjugated primary antibodies were used in this study. Considering that more than 200 primary antibodies were used, the conjugation procedures may be highly different among them, as the concentrations of commercial primary antibodies are different, as are their buffers (e.g., buffers w/ or w/o stabilizer proteins). However, the methods section on the conjugation of primary antibodies and

fluorophores is very simple. Did the authors perform a conjugation optimization for each antibody, such as buffer exchange or degree of labeling optimization? If so, providing such information would be highly beneficial to potential readers who want to use this method.

We apologize for the lack of explanation. We performed conjugation optimization prior to the experiments using an anti-H3.1 antibody, not using all antibodies. Reaction of the antibody with equimolar DBCO-SS-NHS provided the most efficient conjugation without affecting staining. We have added text to explain these details to 'Preparation of PECAbs' in the Materials and Methods section (Line 445 on Page 14).

Also, we purchased BSA-free antibodies to simplify the antibody purification step before labeling. To provide more detail on the antibody labeling, we have added the following text to the Materials and Methods section (Line 411 on Page 13).

Text (Line 445 on Page 14)

Preparation of PECAbs using one-step labeling. For Alexa 488 azide (Thermo Fisher Scientific, A10266) and Alexa 594 azide (Thermo Fisher Scientific, A10270), antibodies were labeled with the fluorophores in a one-step reaction. To prepare NHS-SS-fluorophore couples, DBCO-SS-NHS ester (CONJU-PROBE, CP-2024) and the Alexa Fluor-azide were pre-reacted in DMSO for 3 hours at room temperature. The fluorophore-SS-NHS ester products were stored at $-20\text{ }^{\circ}\text{C}$ until use. Purchased antibodies (100 μg) underwent buffer exchange to $1 \times \text{PBS}$ using 10K Amicon Ultra centrifugal filters (Sigma Aldrich, UFC501096) to remove sodium azide and glycerol. Antibodies were reacted with an equal molar quantity of the fluorophore-SS-NHS ester in 100 mM $\text{NaHCO}_3/\text{PBS}$ for 1 hour at room temperature. The products were purified using 10K Amicon Ultra centrifugal filters (Sigma Aldrich) and stored in 50% glycerol/PBS at $-20\text{ }^{\circ}\text{C}$ until use.

Preparation of PECAbs using two-step labeling. For Janelia fluor 646 azide (Tocris Bioscience, 7088), antibodies were labeled with the fluorophore in two-step reactions. Antibodies, buffer-exchanged as above, were reacted with an equal molar quantity of DBCO-SS-NHS ester in 100 mM $\text{NaHCO}_3/\text{PBS}$ for 1 hour at room temperature. The products were purified using 10K Amicon Ultra centrifugal filters (Sigma Aldrich) to remove unreacted DBCO-SS-NHS ester. The products were made up to 70 μL with PBS and then mixed with 0.7 μL of 10 mM fluor dye azide. After incubation for 24 hours at $4\text{ }^{\circ}\text{C}$ using a rotator, the products were purified using 10K Amicon Ultra centrifugal filters (Sigma Aldrich) and stored in 50% glycerol/PBS at $-20\text{ }^{\circ}\text{C}$ until use.

Text (Line 411 on Page 13)

BSA-free antibodies were purchased to facilitate the buffer exchange step.

6. In their discussion, the authors highlighted the possibility of steric hindrance occurring among antibodies, and they took meticulous steps to optimize the staining order. Specifically, they prioritized the application of antibodies with lower affinities. It would be helpful for prospective users of this technique if the authors could provide details on how they assessed antibody affinity.

We agree to provide the details of how the staining efficiency of PECABs was evaluated. We chose the PECAB staining order according to staining efficiency instead of antibody affinity. The reason for this is that the antigen-specific signal obtained by imaging depends on the fluorescent labeling efficiency of the PECAB. To calculate the staining efficiency, we stained IMR90 cells with PECABs, calculated staining scores as laser power (%) \times exposure time (ms) at which the fluorescent intensity of the specific signal is approximately 200. Sequential staining was then performed with the PECAB staining order rearranged from highest to lowest score. We have added the following text to the Materials and Methods (Line 548 on Page 16) [which is the same response to Reviewer 3 remark 4 (ii)].

Text (Line 551 on Page 16)

The staining order was carefully considered by the staining efficiency of the PECABs. To assess the staining efficiency, IMR90 cells were stained with PECABs, and scores were calculated by the laser power (%) \times exposure time (ms) at which the fluorescence intensity of the specific signal was approximately 200. Then, SeqIS was performed by rearranging the PECAB staining order from highest to lowest.

7. Based on the findings presented in Figure S6A, it is evident that washing with PBS results in an approximately 50% cell loss, while the use of TCEP treatment hardly causes any cell loss. It is not easy to understand how TCEP treatment induces less cell loss than a simple PBS wash.

One explanation for this observation is that disulfide bonds in cell surface membrane proteins are reduced by TCEP, producing thiols, which react with the remaining PFA on the glass to cross-link them. This makes detachment of samples from the glass less likely. However, we could not confirm this possibility because this is a specialized chemistry issue that is beyond our expertise. We also believe it to be beyond the scope of the paper.

8. Line 449 provides multiple erasure solutions. Which solution was used for signal removal in this paper? What are the main differences among these erasing solutions, in terms of erasing performance,

bio-sample integrity, and sample damage?

We apologize for the lack of explanation. TCEP was used for signal removal of PECAb staining. We have rewritten 'Erasing fluorophores' in the Materials and Methods section (Line 492 on Page 15) as below.

Text (Line 495 on Page 15)

Erasing fluorophores

Fluorescent signals of samples stained with PECAbs were erased by incubation with 10 mM TCEP for 30 min. Another two erasing conditions, 3% H₂O₂/20 mM HCl in PBS for CycIF (6) or 3 M urea/3 M guanidinium chloride (GC)/70 mM TCEP in H₂O, pH adjusted to 2.5 with L-glycine for 4i (5) were used to compare sample preservation under the PECAb erasing conditions.

The main differences among these erasing solutions are their effects on sample integrity and damage.

Erasing performance of these erasing solutions depends on the antibody system used. For PECAbs, the SS bond between linker and fluorescent dye is cleaved by 10 mM TCEP. For CycIF (Lin et al., NatComms 2015), fluorescent dye directly conjugated to primary antibodies is inactivated by treating with H₂O₂. For 4i (Gut et al., Science 2018), antibodies used for indirect immunofluorescence are stripped by denaturing agents and cleavage of SS bonds.

The main differences among these erasing solutions for sample integrity and damage are as follows. PECAbs signal can be removed with relatively low concentrations of reducing agents to cleave SS bonds. H₂O₂ used in CycIF causes strong oxidation and alters protein structure. In 4i, 3 M urea/3 M guanidinium chloride (GC)/70 mM TCEP, denatures proteins and cleaves SS bonds using a 7-fold higher concentration of TCEP compared with the PECAb erasing conditions.

Reviewers' Comments:

Reviewer #1:

Remarks to the Author:

The changes made have suitably addressed my earlier concerns. Excellent work!

Reviewer #2:

Remarks to the Author:

The authors have fully addressed my questions and concerns, I agree with the publication of this paper.

Reviewer #3:

Remarks to the Author:

I appreciate the authors' diligent work in the revised manuscript. They have addressed almost all of my concerns, for which I am grateful. However, regarding the authors' response to my first comment on the specificity of the antibodies, I still have concerns regarding the validation of the PECAB antibodies. In the revised version, it appears that the authors stained specimens with PECAB primary antibodies (having fluorophore A), followed by staining the same specimens with a secondary antibody (having fluorophore B) targeting the PECAB primary antibody. If I am mistaken about the experimental procedure, please correct me. If the experiment was conducted in this way, what this result shows is that PECAB primary antibodies can be stained with secondary antibodies, but it does not address the validity of the staining pattern of the PECAB antibodies. What concerns me the most is the staining pattern of the PECAB primary antibody. Given that the authors performed multiplexed imaging of more than 100 antibodies, validating the staining pattern seems to be the most crucial aspect of this work. I think an alternative option would be to display the high-magnification versions of the PECAB antibody staining results shown in Supplementary Figure 1 of the revised manuscript. These images would show the details of the staining pattern of PECAB antibodies, thereby enabling readers to be convinced that the staining pattern of PECAB antibodies corresponds to patterns familiar to them.

REVIEWER COMMENTS

We express our deepest appreciation to the reviewers who reviewed the manuscript and additional comments. We have responded to concern of reviewer 3 in consultation with each of the authors. The comment is addressed below. The reviewers' comments are in blue and our responses are in black.

Reviewer #3 (Remarks to the Author):

I appreciate the authors' diligent work in the revised manuscript. They have addressed almost all of my concerns, for which I am grateful. However, regarding the authors' response to my first comment on the specificity of the antibodies, I still have concerns regarding the validation of the PECAB antibodies. In the revised version, it appears that the authors stained specimens with PECAB primary antibodies (having fluorophore A), followed by staining the same specimens with a secondary antibody (having fluorophore B) targeting the PECAB primary antibody. If I am mistaken about the experimental procedure, please correct me. If the experiment was conducted in this way, what this result shows is that PECAB primary antibodies can be stained with secondary antibodies, but it does not address the validity of the staining pattern of the PECAB antibodies. What concerns me the most is the staining pattern of the PECAB primary antibody. Given that the authors performed multiplexed imaging of more than 100 antibodies, validating the staining pattern seems to be the most crucial aspect of this work. I think an alternative option would be to display the high-magnification versions of the PECAB antibody staining results shown in Supplementary Figure 1 of the revised manuscript. These images would show the details of the staining pattern of PECAB antibodies, thereby enabling readers to be convinced that the staining pattern of PECAB antibodies corresponds to patterns familiar to them.

We appreciate your comment to highlight the specificity of PECAB. We are afraid that we may have misled you by presenting the images at a reduced size, even though the images were obtained by using a 63× lens at NA 1.40, and by failing to explain the details. Therefore, we have added enlarged images in Fig.S1 and added an explanation of the lens detail in the caption by following your suggestion as shown below. We have also shown the representative image in this letter in case these are needed for the review process.

Fig.S1

Caption (Extended data, Line 17 on Page 2)

Fig. S1 Evaluation of PECAbs. Samples (IMR90, A549, and mouse TA tissue) were stained with PECAbs. Images were obtained by using a 63 \times oil objective lens (Plan Apo 1.40 NA). Regions in yellow square are magnified and indicated below for each image. Scale bars = 50 μ m.

Representative images of PECAb staining comparing with standard immunofluorescence shown in the Human Protein Atlas (<https://www.proteinatlas.org>) below.

Reviewers' Comments:

Reviewer #3:

Remarks to the Author:

The authors' revision adequately addressed my comments.